# Preserving Deep Representations in One-Shot Pruning: A Hessian-Free Second-Order Optimization Framework

**Ryan Lucas**[*]    **Rahul Mazumder**[†]

## Abstract

We present SNOWS, a one-shot post-training pruning framework aimed at reducing the cost of vision network inference without retraining. Current leading one-shot pruning methods minimize layer-wise least squares reconstruction error which does not take into account deeper network representations. We propose to optimize a more global reconstruction objective. This objective accounts for nonlinear activations deep in the network to obtain a better proxy for the network loss. This nonlinear objective leads to a more challenging optimization problem—we demonstrate it can be solved efficiently using a specialized second-order optimization framework. A key innovation of our framework is the use of Hessian-free optimization to compute exact Newton descent steps without needing to compute or store the full Hessian matrix. A distinct advantage of SNOWS is that it can be readily applied on top of any sparse mask derived from prior methods, readjusting their weights to preserve deep feature representations. SNOWS obtains state-of-the-art results on various one-shot pruning benchmarks including residual networks and Vision Transformers (ViT/B-16 and ViT/L-16, 86m and 304m parameters respectively). Our open-source implementation is available at https://github.com/mazumder-lab/SNOWS.

## 1 Introduction

Modern deep learning-based vision models, particularly convolutional neural networks (CNNs) (Lecun et al., 1998) and more recent vision transformers (ViTs) (Dosovitskiy et al., 2021), have achieved remarkable performance in various computer vision tasks, including image classification, object detection, and visual tracking (Chai et al., 2021). However, their success comes at the cost of substantial computational and memory resources, and these models are only trending towards increasingly large scales (Dehghani et al., 2023). Neural network pruning, a class of techniques aimed at reducing the number of parameters by selectively removing weights or connections, has emerged as a solution to reduce the inference demands of large-scale vision networks (Goel et al., 2020; He & Xiao, 2023; Kuznedelev et al., 2023), alongside methods such as quantization (Cheng et al., 2024).

Many pruning approaches require an expensive retraining phase following weight removal to recover network performance (Liu et al., 2019; Frankle & Carbin, 2019). Especially as networks scale, practitioners looking to prune and cheaply deploy a model often lack the resources to fully retrain it and may not even have access to the model's dataset or optimizer. While classical *one-shot pruning* approaches (where weights are removed in a single step rather than iteratively) typically included retraining Hagiwara (1994), recent work has explored one-shot pruning without this computationally expensive step (Frantar & Alistarh, 2023; 2022; Singh & Alistarh, 2020).

One-shot pruning approaches typically use second-order approximations of the neural network loss function in their pruning criteria or objective function, an approach dating back to (LeCun et al., 1989; Hassibi & Stork, 1992). Modern second-order approaches fall into two main categories: global methods, which consider the entire network at once, and local methods, which prune one layer at a time. Global methods decide which weights to prune and how to update the remaining

---

[*]MIT Operations Research Center (`ryanlu@mit.edu`)

[†]MIT Sloan School of Management, MIT Operations Research Center, and MIT Center for Statistics (`rahulmaz@mit.edu`)

weights, based on a second-order approximation of the downstream task loss function. However, since computing and inverting the full Hessian matrix is expensive, these methods often replace the Hessian with an approximation such as the Fisher information matrix or further approximations thereof (Singh & Alistarh, 2020; Frantar et al., 2021; Benbaki et al., 2023; Wang et al., 2019). Like retraining approaches, global pruning methods can be difficult to scale to the largest vision networks due to their need to take full network gradients (e.g. in the Fisher computation). Local methods on the other hand, which have become popular for large networks, replace the task loss with a local layer-wise least squares reconstruction loss (Dong et al., 2017a; Frantar & Alistarh, 2022; Meng et al., 2024a), for which the Hessian can be computed in a simple, straightforward fashion. In these local layer-wise reconstruction methods, the only optimization variables are the weights at a single layer, hence layer-wise Hessians can be isolated and the network pruned one layer at a time. These local methods are computationally efficient, but may not provide the most accurate proxy of the loss of the network and neglect the influence of deeper feature representations.

In this paper, we present SNOWS[1], a neural network pruning framework that bridges the gap between these global and local one-shot pruning approaches. Unlike global or retraining-based methods, SNOWS only requires calculating gradients for individual layers. At the same time, the SNOWS formulation goes beyond standard layer-wise reconstruction and optimizes a more global reconstruction objective (cf. Equation 3). The motivation is to preserve not just a local set of features, but to consider the effect of pruning on the more global set of learned representations. While this objective leads to a more challenging nonlinear optimization problem that cannot be solved by standard layer-wise methods, we show that it can solved efficiently using a specialized second-order optimization framework. Unlike prior global approaches that make approximations to the Hessian, SNOWS optimizes an *exact* second-order approximation. A key driver of the scalability of our method is the use of Hessian-free optimization (Martens, 2010), which allows us to efficiently compute and optimize the second-order approximation without explicitly forming or storing the full Hessian matrix. This enables SNOWS to scale to the largest CNNs and ViTs. For instance, using our method the ViT-L/16 model ($\approx 304$m parameters) can be pruned on a single A100 GPU.

Semi-structured pruning approaches have received more attention since they lead to more tangible acceleration on modern computing hardware. We primarily target $N{:}M$ sparsity (Mishra et al., 2021), a semi-structured pruning approach where $N$ out of every $M$ (contiguous, aligned) weights are retained. This sparsity pattern leads to greater acceleration than unstructured sparsity by aligning with hardware indexing patterns, such as those in NVIDIA's Sparse Tensor Cores (Pool & Yu, 2021), and leads to lesser accuracy loss than structured formats (Zhang et al., 2023). Unlike previous methods that require training $N{:}M$ sparse networks from scratch (Zhou et al., 2021; Zhang et al., 2022) to achieve performance competitive with dense networks, SNOWS can prune to $N{:}M$ sparsity in one-shot using just a few thousand samples.

**Contributions.** We make the following contributions to the one-shot pruning problem:

- We introduce a novel post-training pruning framework that operates at the layer-wise level to optimize a more global reconstruction objective to account for nonlinear activations deep in the network. We demonstrate empirically that this approach provides a better proxy for the network loss compared to layer-wise least squares reconstruction methods.

- We develop an efficient second-order optimization framework to solve the challenging nonlinear optimization problem posed by our reconstruction objective. We use Hessian-free optimization integrated with a customized Conjugate Gradient (CG) method to efficiently solve the second-order approximation. Our CG method exploits sparsity in the linear system defined by the second-order approximation to compute exact Newton descent steps in a memory-efficient way, scaling to problems with layer-wise Hessians as large as $4.2\text{m} \times 4.2\text{m}$.

- SNOWS delivers state-of-the-art performance in one-shot pruning for computer vision, achieving superior results on various network architectures and datasets. Our method can prune ResNet50 to $1{:}4$ sparsity with a 1.5% drop in test accuracy on CIFAR-100 and a 9.4% drop on ImageNet-1k using just 3,000 training examples, where prior state-of-the-art one-shot $N{:}M$ pruning methods have a drop of 7.6% and 29.42% respectively. Similarly, for unstructured sparsity, our method can prune MobileNet to 70% sparsity with a 5.6% drop in accuracy, whereas the best available

---

[1]**S**tochastic **N**ewton **O**ptimal **W**eight **S**urgeon (**SNOWS**)

competing method has a 12.55% drop. In addition to improving performance on CNNs, our method scales to modern ViT networks with up to 304m parameters.

## 2 RELATED WORK

**One-shot pruning without retraining.** Recent work has demonstrated the possibility of pruning networks in a single step without requiring expensive retraining Frantar & Alistarh (2023; 2022); Singh & Alistarh (2020). In this setting, one typically assumes access to a limited sample of calibration data (e.g. a few thousand examples) and the goal is to compress the network with minimal accuracy loss relative to the dense model. This modern approach to one-shot pruning builds on classical work that developed second-order approximations of the task loss function. Early approaches (Lecun et al., 1998; Hassibi & Stork, 1992) considered the entire network's weights but were computationally intractable for large models. More recent methods like WoodFisher Singh & Alistarh (2020) and Combinatorial Brain Surgeon (Singh & Alistarh, 2020), as well as CHITA (Benbaki et al., 2023), improve scalability by replacing the Hessian with the Fisher information matrix or using block-diagonal approximations. Other works use Kronecker product approximations of the Fisher matrix (Wang et al., 2019).

These global approaches optimize a more accurate proxy for the network loss function, but often struggle to scale to the largest vision networks, hence layer-wise approaches which focus on *reconstruction* have become a popular alternative. Layer-wise OBS (Dong et al., 2017a) formulates the problem of reconstructing the linear activations of the dense network as a linear regression problem which has a standard least squares Hessian, and show that the overall network reconstruction error is bounded by a quantity depending on the error given at each of the layer-wise problems. The OBC framework (Frantar & Alistarh, 2022) introduces an efficient method for greedy backward elimination on this layer-wise problem using fast low rank updates of the Hessian inverse. There have been limited attempts to incorporate nonlinearity into layer-wise approaches, but several papers have reported the need to consider broader notions of network connectivity as pruning criteria and loss functions (Hoang et al., 2023; He & Zhou, 2024). The Net-Trim framework solves a layer-wise regression problem with ReLU activation (Aghasi et al., 2017), accounting for the non-linearity by reformulating the ReLU using linear inequalities and solving the resulting convex program. In the Net-Trim formulation, the weights are optimized in a way that is "aware" of the subsequent non-linearity by incorporating the effects of the ReLU activation into the pruning loss function. Our objective is similar in spirit, but we make no assumptions about the exact form of the non-linearity, only requiring that the underlying functions are twice-differentiable. Moreover, our formulation goes beyond the activations at a single layer.

**Structured and semi-structured pruning in computer vision.** Unstructured pruning, which involves pruning individual weights in an unrestricted format, leads to high compression ratios but often cannot realize tangible speed-ups except at very high sparsity levels or when run on specialized hardware (Cheng et al., 2024). Hence there has been a movement towards structured pruning methods (He & Xiao, 2023), which lead to actual speed-ups by imposing regularity on the sparsity pattern. In CNNs, structured pruning can involve removing entire filters (Zhou et al., 2016) or specific channels (Meng et al., 2024c). Recently, NVIDIA (Mishra et al., 2021) released $N{:}M$ Sparse Tensor Cores, the current version of which includes a 2:4 sparsity pattern that leads to double the throughput on dense matrix multiplications. Hence there has been several works applying $N{:}M$ pruning in CNNs (Pool & Yu, 2021; Zhou et al., 2021). There has been more limited work on compressing ViT networks, in part due to their scale. Most of the recent work focuses on unstructured sparsity (He & Zhou, 2024). The CAP framework (Kuznedelev et al., 2023) obtains strong results in one-shot unstructured pruning in DeiT networks (Touvron et al., 2021), though they use 4096 gradient evaluations ($\approx 520$k samples) in their one-shot pruning experiments, and use fine-tuning to recover network accuracy. Previous work on structured and semi-structured ViT pruning has taken various approaches. Several works (Zhu et al., 2021; Yu & Wu, 2021) focused on pruning redundant channel dimensions (rows) in the ViT. (Yu et al., 2022a) proposed pruning both the width (number of neurons) and depth (number of layers) of ViTs, which centers on architectural changes rather than specifically weight pruning. Notably, almost all existing work on ViT pruning requires fine-tuning to recover dense model performance, and hence the literature on one-shot structured pruning in VITs is very limited.

## 3 PROBLEM DEFINITION AND BACKGROUND

**Notation.** In a neural network with $L$ layers, the network function is represented as $f(\boldsymbol{X}^0, \boldsymbol{W})$, where $\boldsymbol{W} = (\boldsymbol{W}^1, \boldsymbol{W}^2, \dots, \boldsymbol{W}^L)$ is the collection of weight matrices across layers, and $\boldsymbol{X}^0 \in \mathbb{R}^{n \times d_0}$ is the input, with number of data points $n$ and input dimension $d_0$. For $\ell \geq 1$, we use $\boldsymbol{X}^\ell \in \mathbb{R}^{n \times d_{\ell-1}}$ to refer to the inputs to layer $\ell$ and $\boldsymbol{Y}^\ell \in \mathbb{R}^{n \times d_\ell}$ to refer to the outputs of layer $\ell$; therefore, the input to layer $\ell+1$ is the output of layer $\ell$, or $\boldsymbol{X}^{\ell+1} = \boldsymbol{Y}^\ell$. The weight matrix for the $\ell$-th layer is denoted $\boldsymbol{W}^\ell \in \mathbb{R}^{d_{\ell-1} \times d_\ell}$. The output of layer $\ell$ is computed as $\boldsymbol{Y}^\ell = f^\ell(\boldsymbol{X}^\ell, \boldsymbol{W}^\ell)$, where the function $f^\ell$ denotes the operations at layer $\ell$, which typically consist of an linear transformation followed by a point-wise nonlinear activation, i.e., $f^\ell(\boldsymbol{X}^\ell, \boldsymbol{W}^\ell) = \sigma(\boldsymbol{X}^\ell \boldsymbol{W}^\ell)$, where $\sigma$ is an activation function applied element-wise, though $f^\ell$ can also include much more general operations such as residual connections, pooling operations, etc. For a twice-differentiable loss function $\mathcal{L}$ depending on the network parameters $\boldsymbol{W}$ or a subset, we denote the gradient as $\nabla \mathcal{L}(\boldsymbol{W}) = \partial \mathcal{L}/\partial \boldsymbol{W}$ and the Hessian $\boldsymbol{H} = \partial^2 \mathcal{L}/\partial \boldsymbol{W}^2$.

**Background on Global Model.** Global methods, originating with the Optimal Brain Surgeon (OBS) framework (Lecun et al., 1998; Hassibi & Stork, 1992), minimize the following loss function for neural network compression:

$$\min_{\widehat{\boldsymbol{W}} \in \mathcal{W}} \mathcal{L}(\widehat{\boldsymbol{W}}) = \mathbb{E}_{(\boldsymbol{X}^0, \boldsymbol{Y})} \left[ E(\boldsymbol{Y}, f(\boldsymbol{X}^0, \widehat{\boldsymbol{W}})) \right] \tag{1}$$

where $\boldsymbol{Y}$ are the ground truth labels, $f(\boldsymbol{X}^0, \widehat{\boldsymbol{W}})$ represents the final output of the compressed network, $E(\cdot)$ is the task loss, e.g., cross-entropy for classification, and $\mathcal{W}$ is a predefined set of sparse weight matrices. The canonical example of $\mathcal{W}$ is the set of $S$-sparse matrices, $\mathcal{W} = \{\widehat{\boldsymbol{W}} \mid \|\widehat{\boldsymbol{W}}\|_0 \leq S\}$, though $\mathcal{W}$ can also represent much more general sets such as structured-sparse matrices. The very natural idea here is to compress the network to preserve performance on the downstream task. To solve the problem, OBS uses a second-order approximation to Eqn (1).[2] Intuitively, a second-order method is well-motivated by the importance of interactions in pruning. Time has revealed however that there are practical issues limiting the applicability of the OBS framework. Most importantly, the OBS solution to Eqn (1) requires computing the Hessian of the entire network's weights. For modern deep neural networks, computing the full Hessian matrix is simply not tractable.

**Background on Layer-wise Model.** A more practical alternative to global methods is the popular layer-wise proxy, replacing Eqn (1) with the following regression problem:

$$\min_{\widehat{\boldsymbol{W}}^\ell \in \mathcal{W}} \left\| \boldsymbol{X}^\ell \boldsymbol{W}^\ell - \boldsymbol{X}^\ell \widehat{\boldsymbol{W}}^\ell \right\|_2^2 \tag{2}$$

The goal in Eqn (2) is to preserve the linear activations of the model at a particular layer while enforcing sparsity in $\widehat{\boldsymbol{W}}^\ell$. This formulation, which was introduced in (Dong et al., 2017a), has practical appeal since the only decision variables are the weights at a single layer $\widehat{\boldsymbol{W}}^\ell$, so the Hessian has reduced dimensionality $(d_{\ell-1} \times d_\ell)^2$ compared to that of the full network. Moreover, since the operations on $\widehat{\boldsymbol{W}}^\ell$ are linear, the objective has a standard least squares form, and its Hessian is given by $\boldsymbol{H} = 2\boldsymbol{X}^\ell \boldsymbol{X}^{\ell\top}$, meaning it can be computed in parallel across the row dimension of $\boldsymbol{X}^\ell$.

## 4 THE SNOWS METHOD

**Our Improved Layer-wise Model.** While more tractable than global approaches, the layer-wise formulation in Eqn (2) focuses solely on preserving the linear activations at a particular layer without considering the impact on subsequent layers' representations. In this paper we instead consider the following *layer-wise* model:

$$\min_{\widehat{\boldsymbol{W}}^\ell \in \mathcal{W}} \mathcal{L}(\widehat{\boldsymbol{W}}^\ell) := \sum_{k=0}^{K} \left\| \boldsymbol{Y}^{\ell+k} - f^{\ell:\ell+k}(\boldsymbol{X}^\ell, \widehat{\boldsymbol{W}}^\ell) \right\|_2^2 \tag{3}$$

where $f^{\ell:\ell+k}(\cdot)$ is defined as the composition of the next $k$ operations starting from layer $\ell$:

$$f^{\ell:\ell+k}(\boldsymbol{X}^\ell, \boldsymbol{W}^\ell) = f^{\ell+k}(\cdot, \boldsymbol{W}^{\ell+k}) \circ f^{\ell+k-1}(\cdot, \boldsymbol{W}^{\ell+k-1}) \circ \dots \circ f^\ell(\cdot, \boldsymbol{W}^\ell)$$

---

[2]We omit the details of the second-order approximation for brevity, but provide more in A.1.1

and $\boldsymbol{Y}^{\ell+k} = f^{\ell:\ell+k}(\boldsymbol{X}^{\ell}, \boldsymbol{W}^{\ell})$ are the dense target outputs after the $k$-th operation starting from layer $\ell$. Hence, the loss function considers the effect of $\widehat{\boldsymbol{W}}^{\ell}$ on all layers and operations up to $\ell + K$. A natural motivation for Eqn (3) is that pruning weights at a particular layer affects not only the outputs of that layer but also activations in subsequent layers. Namely, the solution to the layer-wise problem in Eqn (2) does not necessarily minimize the discrepancy in the outputs of deeper layers:

$$\mathrm{argmin}_{\widehat{\boldsymbol{W}}^{\ell}} \left\| \boldsymbol{X}^{\ell}\boldsymbol{W}^{\ell} - \boldsymbol{X}^{\ell}\widehat{\boldsymbol{W}}^{\ell} \right\|_2^2 \neq \mathrm{argmin}_{\widehat{\boldsymbol{W}}^{\ell}} \left\| \boldsymbol{Y}^{\ell+k} - f^{\ell:\ell+k}(\boldsymbol{X}^{\ell}, \widehat{\boldsymbol{W}}^{\ell}) \right\|_2^2$$

for all $k > 0$, where the constituent functions can include non-linear activations, pooling operations, and residual connections, among others. We provide more detail on the motivation for the SNOWS loss function in subsubsection A.1.2.

**Special Cases of Eqn (3).** The loss function is a flexible proxy for the neural network loss function, where the parameter $K$ can be varied depending on available computational resources. When $K = 0$, the loss function reduces to the layer-wise framework. When $K = 1$, the formulation is analogous to Net-Trim for ReLU networks Aghasi et al. (2017), but also generalizes it to other activation functions. Setting $K = K_{\max}$ which denotes the number of operations after layer $\ell$ up to the output layer, extends the optimization to all operations following layer $\ell$. In Figure 1, we observe how the value of $K$ influences network performance when pruning the full ResNet20 and ResNet50 networks to 1:4 sparsity using SNOWS. In Table 3 in subsubsection A.2.4, we provide an ablation on the run-time and peak memory usage when varying $K$.

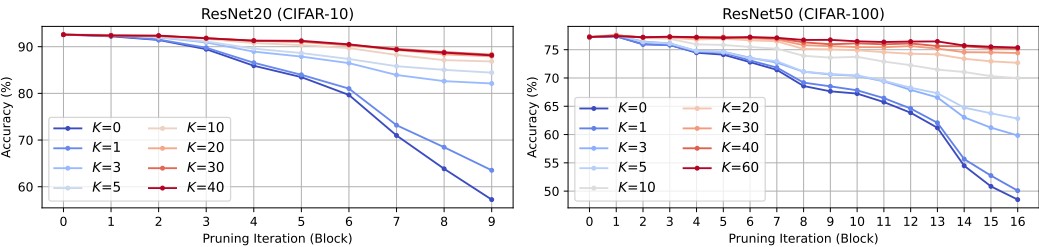

Figure 1: Effect of varying $K$ in the loss function in Eqn (3) on out-of-sample accuracy pruning ResNet20 on CIFAR-10 and ResNet50 on CIFAR-100 to 1:4 sparsity (74% and 66% respectively). Increasing $K$ improves the accuracy of the pruned network, at the cost of higher computation time.

**Cheap layer-wise gradients without computing the full computational graph.** The gradient of Eqn (3) function with respect to the pruned weights $\widehat{\boldsymbol{W}}^{\ell}$ can be expressed as:

$$\nabla\mathcal{L}(\widehat{\boldsymbol{W}}^{\ell}) = \sum_{k=0}^{K} \left[ -2\left(\boldsymbol{Y}^{\ell+k} - f^{\ell:\ell+k}\left(\boldsymbol{X}^{\ell}, \widehat{\boldsymbol{W}}^{\ell}\right)\right) \frac{\partial f^{\ell:\ell+k}\left(\boldsymbol{X}^{\ell}, \widehat{\boldsymbol{W}}^{\ell}\right)}{\partial \widehat{\boldsymbol{W}}^{\ell}} \right] \tag{4}$$

An important advantage of this formulation is that it allows for reduced memory requirements compared to computing full network gradients. Specifically, the gradient has dimensionality $\dim(\boldsymbol{W}^{\ell})$, which is significantly smaller than the total number of parameters in the full network. Moreover, the computational graph required to compute this gradient is shorter because it includes only the operations from layer $\ell$ up to $\ell + K$, rather than the entire network, which minimizes the number of activations that need to be constructed and stored on the computational graph.

Each function $f^{\ell:\ell+k}$ can represent linear transformations and several non-linear operations, so while this is still a layer-wise problem depending only on $\widehat{\boldsymbol{W}}^{\ell}$, it is no longer a linear least squares problem. Hence, the Hessian $\boldsymbol{H}$ is no longer equal to $2\boldsymbol{X}^{\ell}\boldsymbol{X}^{\ell\top}$, and does not separate in general. To form the Hessian we need to compute all second-order partial derivates

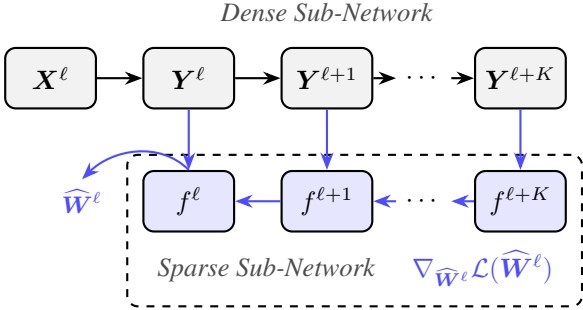

Figure 2: Computional graph to minimize Eqn (3).

of $f^{\ell:\ell+k}$ for each $k = 0, ..., K$ which is also not tractable for large networks. Nonetheless, we now present a tractable framework to optimize Eqn (3).

**An Efficient Second-Order Optimization Framework to Optimize** Eqn (3). Our $K$-step objective induces a discrete, nonlinear optimization problem which cannot necessarily be solved in a similar fashion to previous layer-wise methods (Dong et al., 2017a; Aghasi et al., 2017; Frantar & Alistarh, 2022). Thus, we adopt a similar framework to more global methods by minimizing a local quadratic approximation of our nonlinear objective. However, given the difficulty of forming the Hessian directly, we use Hessian-free optimization (Martens, 2010) to optimize the local quadratic approximation, combined with a CG method that exploits sparsity for efficient updates. We now describe the key components of our framework.

**Hessian-free Optimization.** Given the difficulty of forming the Hessian directly, we opt to use Hessian-free optimization (Martens, 2010) to optimize the local quadratic approximation. Hessian-free optimization is a second-order optimization method that avoids explicit computation of the Hessian matrix by exploiting the fact that while the Hessian matrix $\boldsymbol{H}$ is expensive to compute, the so-called Hessian product $\boldsymbol{H} \cdot \boldsymbol{\delta}$ can be computed as:

$$\boldsymbol{H} \cdot \boldsymbol{\delta} = \lim_{\epsilon \to 0} \frac{\nabla \mathcal{L}(\boldsymbol{W}^\ell + \epsilon \boldsymbol{\delta}) - \nabla \mathcal{L}(\boldsymbol{W}^\ell)}{\epsilon} \tag{5}$$

which has a cost of 1 additional gradient evaluation when computed via finite differences. Note also that $\dim(\boldsymbol{H} \cdot \boldsymbol{\delta}) = \dim(\boldsymbol{W}^\ell)$, meaning that the memory requirements of $\boldsymbol{H} \cdot \boldsymbol{\delta}$ are the same as for $\boldsymbol{W}^\ell$, unlike the Hessian which is usually far too large to fit on GPU memory. Moreover, since we operate on smaller localized sections of the network, from layer $\ell$ to operation $\ell + K$, our formulation is less susceptible to numerical error that can harm Hessian-free optimization in full network training approaches (Martens, 2010). This allows us to compute exact Hessian products as opposed to Fisher or Gauss-Newton approximations used in prior work on matrix-free training and pruning (Martens & Sutskever, 2012; Singh & Alistarh, 2020; Frantar et al., 2021).

**Decoupling the Discrete and Continuous Problems.** In designing our algorithm for scalability to the largest networks, we develop an approach that is agnostic to the mask. Our approach can be integrated with other techniques for mask selection, or can stand alone with simple methods such as magnitude pruning. Our alternative approach for optimizing the local quadratic approximation can be devised by recognizing Eqn (3) as a bi-level optimization problem:

$$\min_{\boldsymbol{Z} \in \mathcal{Z}} \min_{\widehat{\boldsymbol{W}}_{\boldsymbol{Z}}^\ell \in \mathbb{R}^{d_{\ell-1} \times d_\ell}} \mathcal{L}(\widehat{\boldsymbol{W}}_{\boldsymbol{Z}}^\ell) = \sum_{k=0}^{K} \left\| \boldsymbol{Y}^{\ell+k} - f^{\ell:\ell+k} \left( \boldsymbol{X}^\ell, \widehat{\boldsymbol{W}}_{\boldsymbol{Z}}^\ell \odot \boldsymbol{Z} \right) \right\|_2^2 \tag{6}$$

where $\boldsymbol{Z} \in \mathcal{Z}$ is a binary mask imposing the same sparsity pattern as $\mathcal{W}$ and $\boldsymbol{W}_{\boldsymbol{Z}}^\ell$ denotes the weights when the sparsity pattern is fixed. We omit the superscript $\ell$ from $\boldsymbol{Z}$ for simplicity, though the mask is typically chosen and applied layer-wise also. In our case, fixing $\boldsymbol{Z}$ allows us to perform a second-order Taylor expansion around the masked solution for the weights:

$$\mathcal{L}(\widehat{\boldsymbol{W}}_{\boldsymbol{Z}}^\ell + \boldsymbol{\delta}_{\boldsymbol{Z}}) \approx \mathcal{L}(\widehat{\boldsymbol{W}}_{\boldsymbol{Z}}^\ell) + \nabla \mathcal{L}(\widehat{\boldsymbol{W}}_{\boldsymbol{Z}}^\ell)^T \boldsymbol{\delta}_{\boldsymbol{Z}} + \frac{1}{2} \boldsymbol{\delta}_{\boldsymbol{Z}}^T \boldsymbol{H}_{\boldsymbol{Z}} \boldsymbol{\delta}_{\boldsymbol{Z}} \tag{7}$$

In this approximation, we consider perturbing the weights $\widehat{\boldsymbol{W}}_{\boldsymbol{Z}}^\ell$ for a given mask $\boldsymbol{Z}$, which induces sparsity in the solution. We can now design strategies to minimize Eqn (7) which simplifies the problem dramatically compared to solving a coupled discrete and continuous problem.

**Sparsity in the Second-Order Expansion.** The Taylor expansion in Eqn (7) has several key properties related to the sparsity induced by the mask $\boldsymbol{Z}$. First, consider the gradient $\nabla \mathcal{L}(\widehat{\boldsymbol{W}}_{\boldsymbol{Z}}^\ell) = \frac{\partial \mathcal{L}}{\partial \widehat{\boldsymbol{W}}_{\boldsymbol{Z}}^\ell}$. It follows that $(\nabla \mathcal{L})_i = 0$ wherever $\boldsymbol{Z}_i = 0$. Similarly, the Hessian $\boldsymbol{H}_{\boldsymbol{Z}}$ exhibits sparsity, with $(\boldsymbol{H}_{\boldsymbol{Z}})_{ij} = 0$ wherever $\boldsymbol{Z}_i = 0$ or $\boldsymbol{Z}_j = 0$. Therefore, the sparsity pattern of $\boldsymbol{H}_{\boldsymbol{Z}}$ is the square of the sparsity pattern of $\boldsymbol{Z}$. Specifically, if $\boldsymbol{Z}$ has a sparsity level $s \in [0, 1]$, then the Hessian $\boldsymbol{H}_{\boldsymbol{Z}}$ will have a sparsity level $s^2$. Moreover, any row or column $(\boldsymbol{H}_{\boldsymbol{Z}})_i$ in the Hessian corresponding to $\boldsymbol{Z}_i = 0$ will be entirely zero. We exploit this sparsity in $\boldsymbol{H}_{\boldsymbol{Z}}$ to reduce the dimensionality of the system required to minimize Eqn (7). It is important to note that unlike in the global approximation, we cannot assume $\nabla \mathcal{L}(\widehat{\boldsymbol{W}}_{\boldsymbol{Z}}^\ell) = 0$ in Eqn (7). In the global case this is justified since they start by computing the gradient and Hessian of the fully dense network and each time make optimal weight

adjustments. Whereas in our approximation, this would amount to making the assumption that the current weights are optimal for any mask $\boldsymbol{Z}$, even after other weights have been deactivated. Thus the gradient term remains and minimizing Eqn (7) with respect to the weight direction $\boldsymbol{\delta_Z}$ gives:

$$\min_{\boldsymbol{\delta_Z}} \left\{ \boldsymbol{\delta_Z^T} \nabla \mathcal{L}(\widehat{\boldsymbol{W}}_{\boldsymbol{Z}}^\ell) + \frac{1}{2} \boldsymbol{\delta_Z^T} \boldsymbol{H_Z} \boldsymbol{\delta_Z} \right\} \tag{8}$$

Setting the gradient of the above quadratic equal to zero yields the following reduced linear system:

$$\boldsymbol{H_Z} \cdot \boldsymbol{\delta_Z} = -\nabla \mathcal{L}(\widehat{\boldsymbol{W}}_{\boldsymbol{Z}}^\ell) \tag{9}$$

The matrix $\boldsymbol{H_Z} \in \mathbb{R}^{m \times m}$ is the Hessian restricted to the active weights, where $m$ represents the number of weights with $\boldsymbol{Z}_i \neq 0$. Correspondingly, the gradient $\nabla \mathcal{L}(\widehat{\boldsymbol{W}}_{\boldsymbol{Z}}^\ell) \in \mathbb{R}^m$ includes only the components associated with these active weights. As a result, the weight update $\boldsymbol{\delta_Z} \in \mathbb{R}^m$ can be calculated exclusively for the active weights $\boldsymbol{\delta_Z} = -\boldsymbol{H_Z^{-1}} \nabla \mathcal{L}(\widehat{\boldsymbol{W}}_{\boldsymbol{Z}}^\ell)$, which yields the celebrated Newton update (Battiti, 1992), but on a greatly reduced linear system.

**CG Method.** Directly forming and inverting the Hessian $\boldsymbol{H_Z}$, even if sparse, is still not tenable. Here we exploit the fact that the left-hand side Hessian product in Eqn (9) can be computed via Eqn (5), and hence the corresponding linear system can be solved by querying the Hessian product within the CG method (Nocedal & Wright, 2006). To ensure that the Hessian is positive definite, particularly when the objective function is not strongly convex, we add Levenberg-Marquardt regularization to the Newton update. This is a small dampening regularization to the Hessian diagonal, resulting in the following system:

$$(\boldsymbol{H_Z} + \lambda I) \cdot \boldsymbol{\delta_Z} = -\nabla \mathcal{L}(\widehat{\boldsymbol{W}}_{\boldsymbol{Z}}^\ell), \tag{10}$$

This leads to Algorithm 1 to solve for the Newton update, which relies solely on products with the Hessian. Since each call to Eqn (5) requires computing the gradient, the number of iterations in CG is a critical factor in determining the run-time of the overall algorithm. However, although CG can require up to $m$ iterations to fully converge, it often makes significant progress in far fewer steps. In subsubsection A.2.3, we provide an ablation study where we vary the maximum number of steps. CG can typically be safely terminated early after a reasonable number of iterations without significantly degrading the quality of the solution. Given the method for computing the Newton direction, the iterative scheme for performing Newton updates then involves minimizing the quadratic approximation of the loss function at each step to adjust the current solution:

$$\boldsymbol{\delta_Z} \approx \operatorname*{argmin}_{\boldsymbol{\delta_Z}} \left\{ \boldsymbol{\delta_Z^\top} \nabla \mathcal{L}\left(\widehat{\boldsymbol{W}}_{\boldsymbol{Z}}^\ell\right) + \frac{1}{2} \boldsymbol{\delta_Z^\top} \boldsymbol{H_Z} \boldsymbol{\delta_Z} \right\} \qquad \text{(via CG)}$$

$$\widehat{\boldsymbol{W}}_{\boldsymbol{Z}}^\ell = \widehat{\boldsymbol{W}}_{\boldsymbol{Z}}^\ell + \alpha \boldsymbol{\delta_Z} \qquad \text{(Newton Update)}$$

where $\alpha$ is chosen to satisfy the Armijo sufficient decrease condition (Byrd et al., 2016). We provide the full condition in subsubsection A.1.3. In practice, due to hardware memory constraints, we run a batched version of Newton descent called sub-sampled Newton descent. Instead of computing full gradients or Hessian products at once, we sample mini-batches of data to compute stochastic estimates of the gradient and Hessian products. Stochastic Newton descent is a common technique in the literature on second-order optimization methods and has been shown to exhibit strong convergence guarantees (Roosta-Khorasani & Mahoney, 2016). The only alteration we make compared to traditional stochastic Newton algorithms is that we sample Hessian products rather than the Hessian directly. The full algorithm is given in Algorithm 2. We remark that, in practice, the stochastic Newton algorithm typically converges in very few mini-batches. In subsubsection A.2.1 and subsubsection A.2.2, we provide ablations where we compare SNOWS to Stochastic Gradient Descent (SGD) and the Fisher approximation to optimize Eqn (3), finding in both cases that SNOWS obtains better minima and converges faster.

**A complete network pruning algorithm.** So far our descriptions have focused on the method for the layer-wise pruning problem. We now present a recap of the full layer-wise pruning process (Algorithm 2) as well as the algorithm to prune the full network (Algorithm 3). Our approach is flexible and can accommodate any choice of masks $\{Z^1, \ldots, Z^L\}$. Algorithm 3 uses a cascading mechanism for pruning layer by layer, where the next layer's input is adjusted after pruning the current layer. This follows the same core idea as Net-Trim (Aghasi et al., 2017). Unlike a parallel approach, where each layer is pruned independently, this method passes the output of one pruned layer as the input to the next. Specifically, after pruning a layer $\ell$ the input to the next layer is updated as $X^{\ell+1} = f^\ell(X^\ell, \widehat{W}^\ell)$. This ensures that each layer's optimization accounts for changes introduced via pruning in the previous layer.

---

**Algorithm 1** CG Algorithm to Compute $\delta_Z$

---

1: **Input:** Fixed mask $Z$, sparse weights $W_Z^\ell$
2: Compute gradient $g \leftarrow \nabla \mathcal{L}(\widehat{W}_Z^\ell)$
3: Initialize $\delta_Z \leftarrow 0, r \leftarrow -g, p \leftarrow r$
4: **while** $\|r\| \geq$ tol **do**
5: $\quad$ Compute $B(p) = H_Z p + \lambda p$ via Eqn (5)
6: $\quad \eta \leftarrow \frac{r^\top r}{p^\top B(p)}$
7: $\quad$ Update $\delta_Z \leftarrow \delta_Z + \eta p$
8: $\quad r_{\text{new}} \leftarrow r - \eta B(p)$
9: $\quad \gamma \leftarrow \frac{r_{\text{new}}^\top r_{\text{new}}}{r^\top r}$
10: $\quad p \leftarrow r_{\text{new}} + \gamma p$
11: $\quad r \leftarrow r_{\text{new}}$
12: **end while**
13: **return** $\delta_Z$

---

---

| **Algorithm 2** SNOWS: Layer-wise Optimization | **Algorithm 3** SNOWS: Full Network Algorithm |
|---|---|

---

1: **Input:** Layer input $X^\ell$, Dense weights $W^\ell$, Sparse mask $Z^\ell$
2: **Initialize:**
$$\widehat{W}^\ell \leftarrow W^\ell \odot Z^\ell$$
3: **for** $b = 1, ..., B$ **do** ($\triangleright$ *mini-batches*)
4: $\quad$ **Compute $\delta_{Z^\ell}^{(b)}$ via Algorithm 1:**
$$\delta_{Z^\ell}^{(b)} \approx \underset{\delta}{\operatorname{argmin}} \left( \nabla \mathcal{L}^{(b)\top} \delta + \frac{1}{2} \delta^\top H_{Z^\ell}^{(b)} \delta \right)$$
5: $\quad$ **Update weights on the active set:**
$$\widehat{W}_{Z^\ell}^\ell \leftarrow \widehat{W}_{Z^\ell}^\ell + \alpha^{(b)} \delta_{Z^\ell}^{(b)}$$
6: **end for**
7: **Output:** Pruned weights $\widehat{W}^\ell$

**Input:** Network input $X^0$, dense weights $\{W^1, \ldots, W^L\}$, sparse masks $\{Z^1, \ldots, Z^L\}$, layers $L$, horizon parameter $K$
**for** $\ell = 0, \ldots, L$ **do**
$\quad$ **Find $\widehat{W}^\ell$ as:**
$$\underset{\widehat{W}^\ell}{\operatorname{argmin}} \sum_{k=0}^{K(\ell)} \left\| Y^{\ell+k} - f^{\ell:\ell+k}(X^\ell, \widehat{W}^\ell \odot Z^\ell) \right\|_2^2$$
via Algorithm 2 with $K(\ell) = \min(K, K_{\max})$.
$\quad$ **Cascade inputs to the next layer:**
$$X^{\ell+1} = f^\ell(X^\ell, \widehat{W}^\ell)$$
**end for**
**Output:** Pruned weights $\{\widehat{W}^1, \ldots, \widehat{W}^L\}$

---

## 5 EXPERIMENTS ON CNNS AND VISION TRANSFORMERS

**CNN Pruning.** CNNs are a fundamental architecture for computer vision tasks. In a CNN, the weight tensor $W^\ell$ has dimensions $(d_{\text{out}} \times d_{\text{in}} \times k_h \times k_w)$. Here, $d_{\text{out}}$ is the number of output channels (filters), $d_{\text{in}}$ is the number of input channels, and $k_h$ and $k_w$ are the kernel height and width, respectively. We focus on a semi-structured sparsity pattern that has become more popular in recent years, called $N{:}M$ sparsity. This sparsity pattern requires that, for every (non-overlapping) block of $M$ weights, exactly $N$ weights are non-zero. For CNNs, the $N{:}M$ sparsity format requires that the weights be sparse along the input dimension $d_{\text{in}}$ (Pool & Yu, 2021). We follow this convention throughout our experiments. We provide an example of an $N{:}M$ sparse convolutional kernel in Figure 11 in subsection A.3.1 and additional details on how we handled the $K$-step problem.

**Experimental setup.** We evaluate SNOWS across several standard CNN benchmarks. We use the CIFAR-10, CIFAR-100, and ImageNet-1k datasets (Krizhevsky & Hinton, 2009; Deng et al., 2009), where we evaluate performance pruning the ResNet20 and ResNet50 architectures (He et al., 2015) to $N{:}M$ sparsity. Additionally, for experiments on unstructured sparsity, we use the MobileNetV1 architecture (Howard et al., 2017) to compare to previous studies on unstructured pruning. Due to the size of the original ImageNet dataset and the fact that one-shot methods require just a few thousand samples, we use the MiniImagenet-1k dataset, a sample from the ImageNet-1k dataset. We use 10,000 samples from this dataset for evaluation, with 3,000 calibration samples being used

for the $N{:}M$ sparsity experiments. For unstructured sparsity, we use 1,000 calibration samples to be consistent with the methods we compare with (Benbaki et al., 2023).

**Results.** *$N{:}M$ sparsity.* We first evaluate the impact of running SNOWS on top of existing mask selection techniques, OBC (Frantar & Alistarh, 2022), SparseGPT (Frantar & Alistarh, 2023), and MP, that support $N{:}M$ sparsity. Adding SNOWS on top of other pruning techniques consistently improves performance across different methods and datasets, as shown in Figure 1. This is evident in more challenging datasets like CIFAR-100 and ImageNet-1k, where other methods alone perform poorly, but the addition of SNOWS recovers close to the performance of the dense model. For 2:4 sparsity, the choice of mask selection technique has less of an impact on the results. Once SNOWS is applied, the methods achieve similar performance. However, at higher sparsity (1:4), more intensive mask selection techniques, particularly OBC, combined with SNOWS give the best results. We provide latency performance estimates for the $N{:}M$ patterns shown in Figure 1 in subsubsection A.3.5. We also report hyperparameters for SNOWS, including the value of $K$, the dampening factor $\lambda$, and the CG tolerance in subsubsection A.3.6.

*Unstructured Sparsity.* We also compare SNOWS against several popular one-shot pruning methods for unstructured sparsity, including Magnitude Pruning (MP) Gupta et al. (2024), WoodFisher (Singh & Alistarh, 2020), Combinatorial Brain Surgeon (Yu et al., 2022b), and CHITA (Benbaki et al., 2023), using the reported results from the CHITA paper for the competing methods. We remark that our base ResNet20 and MobileNetV1 models start from slightly different starting accuracies, thus we instead show the accuracy change compared to the baseline model in Figure 3 and provide the raw numbers in subsubsection A.3.3. Other one-shot methods optimize both the mask and weights, whereas SNOWS optimizes from a mask obtained through magnitude pruning. Despite this, SNOWS consistently outperforms existing one-shot approaches at all sparsity levels. In subsubsection A.3.4, we provide additional retraining experiments for SNOWS and the best competing method in Figure 3, FALCON Meng et al. (2024b), showing SNOWS can further improve with some additional fine-tuning. We also provide additional run-time comparisons in subsubsection A.3.7.

| Method | CIFAR-10 (ResNet20) | | CIFAR-100 (ResNet50) | | ImageNet-1k (ResNet50) | |
|---|---|---|---|---|---|---|
| | **1:4** | **2:4** | **1:4** | **2:4** | **1:4** | **2:4** |
| Dense Model | 92.58 | | 77.40 | | 75.71 | |
| OBC | 76.77 | 90.98 | 69.79 | 77.03 | 46.29 | 73.74 |
| +SNOWS | **89.09** | **91.86** | **76.05** | **77.11** | **66.33** | **73.87** |
| SparseGPT | 11.06 | 86.05 | 8.27 | 74.10 | 1.61 | 69.46 |
| +SNOWS | **86.10** | **91.94** | **73.66** | **77.10** | **49.32** | **73.59** |
| MP | 10.99 | 63.45 | 9.45 | 70.76 | — | 63.45 |
| +SNOWS | **87.82** | **91.88** | **75.87** | **77.24** | **62.75** | **73.97** |

Table 1: Top-1 Test Accuracy integrating SNOWS with other popular mask selection algorithms for $N{:}M$ pruning.

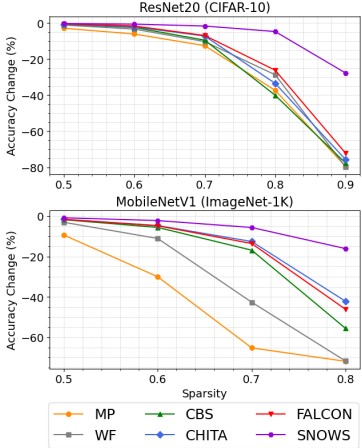

Figure 3: Comparing SNOWS to one-shot pruning methods for unstructured sparsity.

**ViT pruning.** ViTs (Dosovitskiy et al., 2021) have become popular in computer vision by applying the Transformer architecture, originally designed for natural language processing, to image recognition tasks. In ViTs, an image is divided into a sequence of patches and embedded into a vector. These embeddings are then processed through blocks that consist of Multi-Head Self-Attention (MHSA) layers and Feed-Forward Neural Networks (FFNs). (Chen et al., 2022) introduce a framework to make the intermediate attention outputs sparse, while leaving the weights for the query, key, and value as dense matrices. They justify this saying it is particularly hard to prune these weights prior to the attention operations, since it is not the raw weight values that are important, but their interactions within the attention mechanism. In contrast, we jointly optimize over these weights and across all heads, and hence we inherently account for the interactions between the pruned weights and how they influence the attention output. The detailed formulation is provided in subsubsection A.3.3. We also discuss how to prune the output projection module of the MHSA block and MLP layers within the FFN modules in the ViT, which are more standard formulations than for MHSA.

**Experimental setup.** We evaluate SNOWS on ViT-B/16 and ViT-L/16 models, pruning them to 2:4 sparsity on MiniImageNet-1k. Since methods for mask selection in $N$:$M$ sparsity do not extend easily to vision transformers, we employ MP to obtain the 2:4 masks.

**Results.** As shown in Table 2, SNOWS consistently outperforms MP, with higher Top-1 and Top-5 accuracies. When pruning the QKV, output projection, and MLP layers, SNOWS achieves a Top-1 accuracy of 76.57% on ViT/B-16, significantly better than 70.89% for MP. In Figure 4, we plot the attention maps generated by the last attention layer of the dense model, the model pruned with MP and the model obtained when SNOWS is applied on top of MP. SNOWS better preserves the attention patterns in the dense model since it optimizes to preserve the activations via the reconstruction loss. In Figure 5, we show this leads to superior transfer learning performance, where fine-tuning the classifier layer of a ViT pruned with SNOWS outperforms MP when transferring to smaller datasets.

| Original | Dense Model | MP | MP+SNOWS |
|:---:|:---:|:---:|:---:|

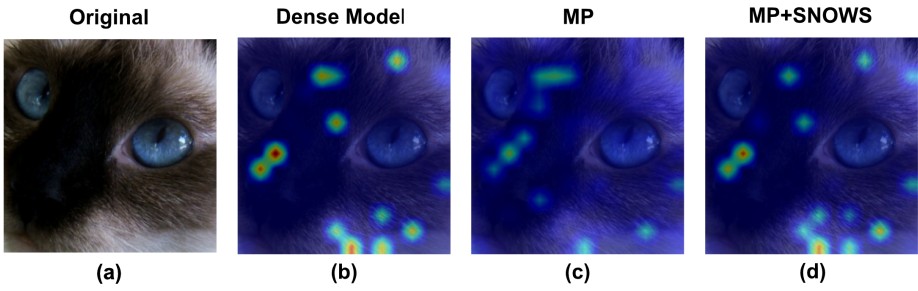

| (a) | (b) | (c) | (d) |
|:---:|:---:|:---:|:---:|

Figure 4: Visualizing (a) the original test image and attention maps from the last layer of (b) the dense VIT/B-16 model, (c) the model obtained by applying a 2:4 MP mask, and (d) the model after applying SNOWS on top of MP. SNOWS optimizes to reconstruct learned activations, preserving features learned by the dense network even in the deepest layers.

| Model | Pruned Layers | Top-1 |
|---|---|---|
| VIT/B-16 Dense: 80.42% | QKV (SNOWS) | **79.45** |
| | QKV (MP) | 78.97 |
| | QKV, Out, MLP (SNOWS) | **76.57** |
| | QKV, Out, MLP (MP) | 70.89 |
| VIT/L-16 Dense: 84.17% | QKV (SNOWS) | **81.01** |
| | QKV (MP) | 75.55 |
| | MLP (SNOWS) | **69.71** |
| | MLP (MP) | 0.28 |

Table 2: Pruning VIT/B-16 and VIT-L-16 to 2:4 sparsity on ImageNet-1k in one shot. SNOWS uses the 2:4 mask obtained from MP. Note that no retraining is involved. We use $n = 5,000$ calibration samples for VIT-B/16 and $n = 20,000$ for VIT-L/16.

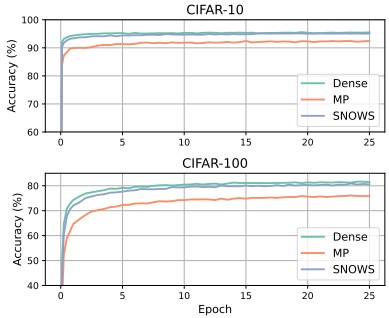

Figure 5: Test accuracy fine-tuning the classifier layer of the 2:4 sparse VIT/B-16 on CIFAR-10 and CIFAR-100.

## 6 LIMITATIONS AND CONCLUSIONS

This work proposes a new layer-wise pruning framework, bridging between the local layer-wise and global pruning loss functions introduced in prior work. The objective of our modified loss function induces a harder non-linear optimization problem compared to the standard layer-wise formulation. We propose an efficient Hessian-free method for optimizing this objective, and demonstrate its efficiency on large-scale networks. There are several limitations to the proposed work: firstly, SNOWS does not do mask selection. This makes it flexible since it can integrate with existing techniques for mask selection, but using discrete optimization as is done in (Benbaki et al., 2023; Meng et al., 2024b;c) on the SNOWS loss function may lead to stronger results. Moreover, the method has little dependence on the computational modules involved in the reconstruction loss. As such, the framework could readily be extended to Text-to-Speech or Large Language Models. However, this would require specializing the formulation of the $K$-step problem as we did for CNN and ViT modules.

## 7 ACKNOWLEDGEMENTS

This research is supported in part by grants from the Office of Naval Research (N000142112841 and N000142212665). We acknowledge the MIT SuperCloud and Lincoln Laboratory Supercomputing Center for providing HPC resources that have contributed to the research results reported within this paper.

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

## A  APPENDIX

### A.1  PRIMARY METHODS AND ANALYSIS

#### A.1.1  BACKGROUND ON OBS AND SECOND-ORDER APPROXIMATIONS.

For pruning the network to achieve the sparsity target, the proposed OBS solution is to start from the dense weights $\boldsymbol{W}$ and minimize a second-order Taylor expansion of the loss function with respect to a perturbation of the weights:

$$\min_{\boldsymbol{\delta}} \mathcal{L}(\boldsymbol{W} + \boldsymbol{\delta}) \approx \mathcal{L}(\boldsymbol{W}) + \nabla \mathcal{L}(\boldsymbol{W})^T \boldsymbol{\delta} + \frac{1}{2} \boldsymbol{\delta}^T \cdot \boldsymbol{H} \cdot \boldsymbol{\delta} \tag{11}$$

subject to the constraint that $\boldsymbol{\delta}_q + \boldsymbol{W}_q = 0$, i.e. a single weight $\boldsymbol{W}_q$ is to be removed, with $\nabla \mathcal{L}(\boldsymbol{W}) = \partial \mathcal{L}/\partial \boldsymbol{W}$ being the gradient and $\boldsymbol{H} = \partial^2 \mathcal{L}/\partial \boldsymbol{W}^2$ being the Hessian matrix. The OBS procedure minimizes this approximation iteratively with respect to the weight perturbation $\boldsymbol{\delta}$, each time obeying the constraint that a single weight is removed. Assuming the dense network is fully trained, they argue $\nabla \mathcal{L}(\boldsymbol{W}) = 0$, which leaves just the second-order term involving $\boldsymbol{\delta}$ in the approximation. Minimizing the resulting constrained formulation with respect to $\boldsymbol{\delta}$ using Lagrange multipliers yields exact saliency scores for each weight $\boldsymbol{W}_q$, given as $\rho_q = -\frac{\boldsymbol{W}_q}{[\boldsymbol{H}^{-1}]_{qq}} \boldsymbol{H}^{-1} \boldsymbol{e}_q$ as well as a method for optimally readjusting the remaining weights $\boldsymbol{\delta} = -\frac{\boldsymbol{W}_q}{[\boldsymbol{H}^{-1}]_{qq}} \boldsymbol{H}^{-1} \boldsymbol{e}_q$, where $\boldsymbol{e}_q$ is the standard basis with all zeros except the $q$-th element which is one. Here, $\rho_q$ is a solution to the discrete problem (a ranking among all weights) while $\boldsymbol{\delta}$ is a solution to the continous problem (an optimal weight readjustment for weights other than $q$, when $q$ is removed).

### A.1.2 MOTIVATION FOR THE SNOWS LOSS FUNCTION

The original motivation of the layer-wise reconstruction objective given in Dong et al. (2017b) was to minimize the impact of pruning at each layer on the final network output. They show that the error $\tilde{\varepsilon}^L = \frac{1}{\sqrt{n}} ||\boldsymbol{Y}^L - \widehat{\boldsymbol{Y}}^L||_F$ of the entire pruned network at the final layer $L$ is bounded as:

$$\tilde{\varepsilon}^L \leq \sum_{\ell=1}^{L-1} \left( \prod_{r=\ell+1}^{L} \left\| \widehat{\boldsymbol{W}}^r \right\|_F \sqrt{\delta E^\ell} \right) + \sqrt{\delta E^L}$$

where $\delta E^\ell$ is the layer-wise error between the outputs at layer $\ell$, i.e., $\delta E^\ell = \frac{1}{n} \left\| \boldsymbol{X}^\ell \boldsymbol{W}^\ell - \widehat{\boldsymbol{X}}^\ell \widehat{\boldsymbol{W}}^\ell \right\|_F^2$. In other words, the difference between the final output of the dense and pruned network is bounded by a product of the layer-wise errors. This means if the layer-wise errors can be kept small, the overall error should be controlled. However, there is an interesting subtlety to how the errors propagate. Say we are pruning layer $\ell$. It would seem as though the contribution by layer $\ell$ to $\tilde{\varepsilon}^L$ is confined to $\delta E^\ell$. However, notice that

$$\widehat{\boldsymbol{X}}^{\ell+1} = \widehat{\boldsymbol{Y}}^\ell = f^\ell(\widehat{\boldsymbol{X}}^\ell, \widehat{\boldsymbol{W}}^\ell)$$

So pruning at the current layer changes the input at the next layer and moreover at all subsequent layers since $\widehat{\boldsymbol{X}}^{\ell+k} = \widehat{\boldsymbol{Y}}^{\ell+k-1} = f^{\ell+k-1}(\widehat{\boldsymbol{X}}^{\ell+k-1}, \widehat{\boldsymbol{W}}^{\ell+k-1})$, for all $k = 1, \ldots, L - \ell$.

So after pruning layer $\ell$, we can observe the expression for the error in the subsequent layers

$$\delta E^{\ell+k} = \frac{1}{n} \left\| \boldsymbol{X}^{\ell+k} \boldsymbol{W}^{\ell+k} - \widehat{\boldsymbol{X}}^{\ell+k} \widehat{\boldsymbol{W}}^{\ell+k} \right\|_F^2$$

At the time of pruning layer $\ell$, $\boldsymbol{W}^{\ell+k} = \widehat{\boldsymbol{W}}^{\ell+k}$, since these weights have not yet been pruned, but notice that already $\boldsymbol{X}^{\ell+k} \neq \widehat{\boldsymbol{X}}^{\ell+k}$, since our pruning decision at layer $\ell$ has affected all layers after it. The error term for the future layers becomes

$$\delta E^{\ell+k} = \frac{1}{n} \left\| (\boldsymbol{X}^{\ell+k} - \widehat{\boldsymbol{X}}^{\ell+k}) \boldsymbol{W}^{\ell+k} \right\|_F^2$$

Hence the error at the future layers is determined by the magnitude of the difference $\boldsymbol{X}^{\ell+k} - \widehat{\boldsymbol{X}}^{\ell+k}$ induced by pruning layer $\ell$. Our formulation, by minimizing $\sum_{k=0}^{K} ||\boldsymbol{Y}^{\ell+k} - \widehat{\boldsymbol{Y}}^{\ell+k}||_F^2 = \sum_{k=0}^{K} ||\boldsymbol{X}^{\ell+k+1} - \widehat{\boldsymbol{X}}^{\ell+k+1}||_F^2$ can be interpreted as directly minimizing the subsequent error terms $\sum_{k=0}^{K} \delta E^{\ell+k}$ during the pruning of layer $\ell$ as opposed to only $\delta E^\ell$.

### A.1.3 ARMIJO SUFFICIENT DECREASE CONDITION.

In the Newton descent procedure, we use the Armijo decrease condition to determine an appropriate step size $\alpha$ that ensures a sufficient decrease in the loss function:

$$\alpha = \operatorname*{argmax}_{\alpha \leq 1} \alpha \quad \text{s.t.} \quad \mathcal{L}\left(\widehat{\boldsymbol{W}}_{\boldsymbol{Z}}^{\ell} + \alpha \boldsymbol{\delta_Z}\right) \leq \mathcal{L}\left(\widehat{\boldsymbol{W}}_{\boldsymbol{Z}}^{\ell}\right) + \alpha \beta \, \boldsymbol{\delta_Z}^{\top} \nabla \mathcal{L}\left(\widehat{\boldsymbol{W}}_{\boldsymbol{Z}}^{\ell}\right),$$

The parameter $\beta$ is a small constant. We use $\beta = 1 \times 10^{-5}$.

### A.1.4 A TOY EXAMPLE SHOWING THE ADVANTAGE OF NEWTON'S METHOD OVER SGD UNDER VARYING CURVATURE.

Consider the following toy problem:

$$\min_{\boldsymbol{w}} \mathcal{L}(\boldsymbol{w}) = \frac{1}{2} \left(\lambda_1 w_1^2 + \lambda_2 w_2^2\right) \tag{12}$$

where $\boldsymbol{w} = [w_1, w_2]^T$ is the weight vector and $\lambda_1, \lambda_2 > 0$ are constants representing sensitivity of the loss (curvature) along the $w_1$ and $w_2$ axes, respectively. Now consider a situation where $\lambda_1, \lambda_2$ are very different. For our example, we choose $\lambda_1 = 1$ (low curvature along $w_1$) and $\lambda_2 = 100$ (high curvature along $w_2$). The condition number of the problem is $\kappa = \lambda_{\max}/\lambda_{\min} = \lambda_2/\lambda_1 = 100$. This creates an ill-conditioned problem with a significant difference in curvature between the two directions. Now compare the convergence behavior for Newton and SGD. For SGD the update is:

$$\boldsymbol{w}^{(k+1)} = \boldsymbol{w}^{(k)} - \eta \nabla \mathcal{L}(\boldsymbol{w}^{(k)}),$$

where $\eta$ is the learning rate and $\nabla \mathcal{L}(\boldsymbol{w}) = \begin{bmatrix} \lambda_1 w_1 \\ \lambda_2 w_2 \end{bmatrix}$. The updates can be written as:

$$\begin{cases} w_1^{(k+1)} = w_1^{(k)} - \eta \lambda_1 w_1^{(k)} = (1 - \eta \lambda_1) w_1^{(k)} \\ w_2^{(k+1)} = w_2^{(k)} - \eta \lambda_2 w_2^{(k)} = (1 - \eta \lambda_2) w_2^{(k)} \end{cases}$$

To ensure convergence, the learning rate needs to satisfy $0 < \eta < \frac{2}{\lambda_{\max}}$ e.g. see (Cohen et al., 2022) for a detailed convergence analysis. With $\lambda_{\max} = \lambda_2 = 100$, a natural choice is $\eta = 0.01$. This implies:

$$w_1^{(k+1)} = (1 - \eta \lambda_1) w_1^{(k)} = (1 - 0.01 \times 1) w_1^{(k)} = 0.99 w_1^{(k)}.$$

This means $w_1$ decreases by 1% per iteration, leading to slow convergence along this direction due to the low curvature. Namely, rewriting the update formula:

$$w_1^{(k)} = (0.99)^k w_1^{(0)}.$$

The optimal solution is clearly $w_1 = 0$. Assuming $w_1^{(0)} = 1$, we can find the number of iterations $k$ required for $w_1^{(k)}$ to be less than a small threshold e.g. $\epsilon = 1e^{-6}$:

$$w_1^{(k)} \leq \epsilon \implies k \geq \frac{\ln(10^{-6})}{\ln(0.99)} \approx 1,375 \text{ iterations.}$$

In contrast, along $w_2$:

$$w_2^{(k+1)} = (1 - \eta \lambda_2) w_2^{(k)} = (1 - 0.01 \times 100) w_2^{(k)} = 0 w_2^{(k)} = 0.$$

Here, $w_2$ converges to the optimum in one iteration. Thus, importantly overall convergence is dominated by the slowest direction (low curvature along $w_1$). Now consider Newton's method on the same problem:

$$\boldsymbol{w}^{(k+1)} = \boldsymbol{w}^{(k)} - \boldsymbol{H}^{-1} \nabla \mathcal{L}(\boldsymbol{w}^{(k)}),$$

where the Hessian is $\boldsymbol{H} = \nabla^2 \mathcal{L}(\boldsymbol{w}) = \begin{bmatrix} \lambda_1 & 0 \\ 0 & \lambda_2 \end{bmatrix}$ and its inverse is: $\boldsymbol{H}^{-1} = \begin{bmatrix} \frac{1}{\lambda_1} & 0 \\ 0 & \frac{1}{\lambda_2} \end{bmatrix}$

The updates become:

$$\begin{cases} w_1^{(k+1)} = w_1^{(k)} - \frac{1}{\lambda_1} \lambda_1 w_1^{(k)} = 0, \\ w_2^{(k+1)} = w_2^{(k)} - \frac{1}{\lambda_2} \lambda_2 w_2^{(k)} = 0. \end{cases}$$

Thus both $w_1$ and $w_2$ reach the minimum in one iteration, regardless of the condition number $\kappa$. Newton's method overcomes the issue of varying curvature by adjusting its updates according to the inverse of the Hessian. Rescaling the gradient components means it takes appropriate sized steps in each direction, i.e. larger steps in low-curvature directions and smaller steps in high-curvature ones. This results in convergence to the minimum in a single iteration for both directions, regardless of the condition number $\kappa$. The trajectories are shown in Figure 6.

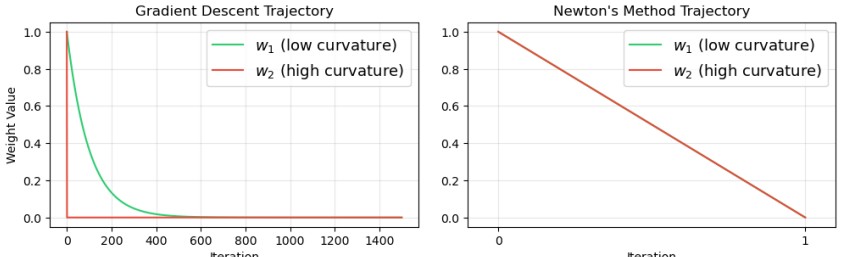

Figure 6: Comparison of convergence rates between SGD and Newton's method on the toy optimization problem in Eqn (12) with highly varying curvature ($\lambda_1 = 1$, $\lambda_2 = 100$). While Newton's method achieves convergence in a single iteration, SGD requires approximately 1,375 iterations due to the high condition number $\kappa = 100$.

## A.2  ABLATION STUDIES

### A.2.1  COMPARING SGD VERSUS NEWTON'S METHOD TO OPTIMIZE EQN (3).

In Figure 7, we compare the time required to optimize the first layer of ResNet20 on CIFAR-10 and ResNet50 on ImageNet-1k, both using a 2:4 mask $\boldsymbol{Z}$ obtained via magnitude pruning. The Stochastic Newton algorithm, when applied to optimize $\widehat{\boldsymbol{W}}_{\boldsymbol{Z}}$, makes significantly faster progress compared to SGD, especially as problem size increases. In the section that follows we provide detailed loss trajectories for both methods on this same problem. Our results indicate that Newton's method moves much farther from the starting solution with each step, whereas SGD progresses much more slowly and generally does not move far from the starting solution. We also provide a more detailed discussion on possible reasons why SGD converges slowly in this setting.

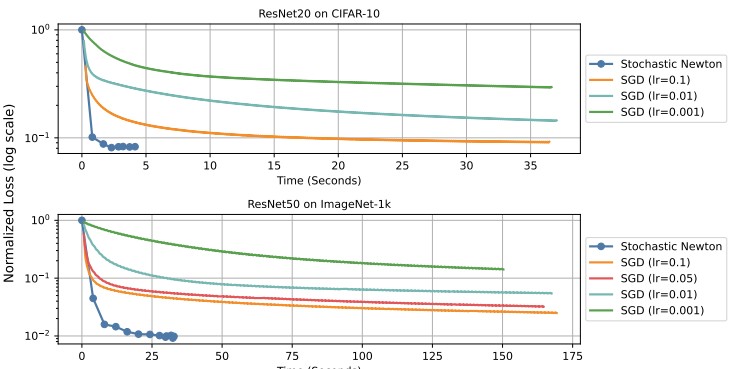

Figure 7: Comparing Newton's method and SGD with minimizing Eqn (3) with $K = 10$, for the first layer in the model. The solution starts with a 2:4 sparse mask obtained via magnitude pruning. Newton's method converges very quickly and requires less time to solve the $K$-step reconstruction problem. SGD is trained for 2000 steps whereas Newton converges in less than 10.

The empirical success of SGD in training neural networks from scratch has been very well-demonstrated (Hardt et al., 2016). Several papers have remarked that this success seems to owe in part to the effect of overparameterization (Neyshabur et al., 2018), where neural networks are trained with more parameters than are necessary for a given task. At the same time, empirical work on sparse fine-tuning has shown that fine-tuning a sparse network with SGD often *does not* lead to good performance (Renda et al., 2020). Other work has found that training a sparse network from scratch with no knowledge of a good initialization is likewise difficult (Evci et al., 2020). Our empirical findings in Figure 7 corroborate this fact, suggesting SGD performs poorly and converges slowly on the optimization problem defined in Eqn (6). In Figure 8 show the weight trajectories for the same problem, finding that SGD does not move far from the starting solution, even after thousands of iterations. We hypothesize that SGD may perform poorly in the sparse setting due to uneven curvature (loss sensitivity) in the coordinates, which is also present in the overparameterized setting but may become exaggerated by the lack of redundancy in sparse networks. In the next section, we show a small toy example comparing SGD and Newton's method for a two-dimensional quadratic optimization problem where the weight directions have varying curvature. The intuition is that SGD must choose a learning rate no larger than the largest eigenvalue of the Hessian since if it is larger, then loss will diverge on that direction. In overparameterized problems, it may be possible to get away with having a long-tail of low curvature directions that take more time to converge. However, in sparse models redundancy is significantly reduced, and this means the optimizer must be more sensitive to each direction. In such cases, choosing a learning rate that works well for all directions becomes increasingly difficult, as there are fewer parameters to absorb the mismatch between the learning rate and the curvature.

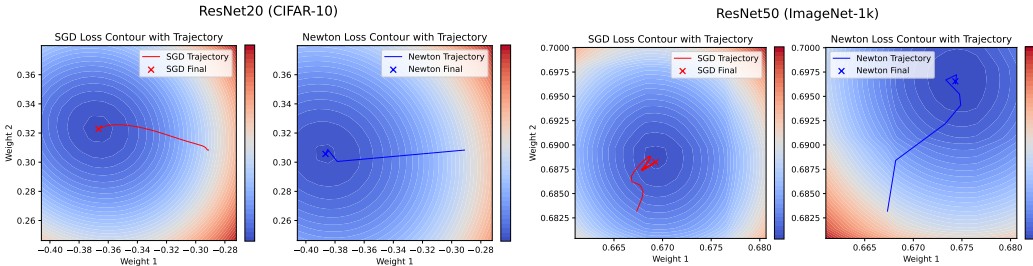

(a) ResNet20: Loss surface for SGD (left) and Newton (right).

(b) ResNet50: Loss surface for SGD (left) and Newton (right).

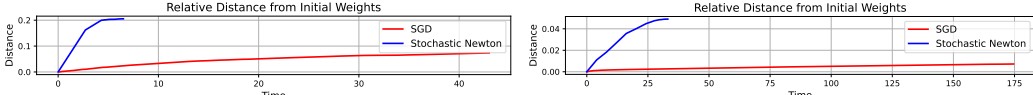

(c) ResNet20: Relative distance between full weight matrix and starting weights.

(d) ResNet50: Relative distance between full weight matrix and starting weights.

Figure 8: Optimization of the first layer for both ResNet20 and ResNet50 using SGD and stochastic Newton from Figure 7. **Top row:** Loss trajectories for 2000 iterations of SGD versus respectively 6 (CIFAR-10) and 13 (ImageNet-1k) iterations of stochastic Newton for the two largest starting weights. **Bottom row:** Relative distance between the full weight matrix and the starting weight matrix for both methods. The relative distance is defined as $\text{dist}(\boldsymbol{W}_t) = \frac{\|\boldsymbol{W}_t - \boldsymbol{W}_0\|^2}{\|\boldsymbol{W}_0\|^2}$ where $\boldsymbol{W}_t$ are the weights at iteration $t$ and $\boldsymbol{W}_0$ are the initial weights.

### A.2.2 USING TRUE HESSIAN VERSUS FISHER APPROXIMATION TO OPTIMIZE EQN (3)

We also observe a significant performance improvement when using the true Hessian over the Fisher approximation. Specifically, we replace the Hessian in Eqn 7 with the Fisher approximation $F = \frac{1}{N} \sum_{i=1}^{N} \nabla_{\widehat{\boldsymbol{W}_Z}} \mathcal{L}_i(\widehat{\boldsymbol{W}_Z}) \nabla_{\widehat{\boldsymbol{W}_Z}} \mathcal{L}_i(\widehat{\boldsymbol{W}_Z})^\top$ and employ the updates $\boldsymbol{\delta}_Z = -F^{-1} \nabla \mathcal{L}(\widehat{\boldsymbol{W}_Z})$. The main advantage of employing the true Hessian via the Hessian-free approach is the accelerated speed of convergence. As illustrated in Figure 9, we plot the objective value when using the Fisher approximation versus the exact Hessian-free steps for all layers in ResNet20 and the first three layers in ResNet50. The results indicate that using Newton's method instead of the Fisher approximation leads to much faster convergence, not only in terms of iterations but also in overall runtime. For the larger ResNet50 networks, running the Fisher approximation to convergence is actually prohibitive, and can take almost half an hour per layer for even the first layers of ResNet50.

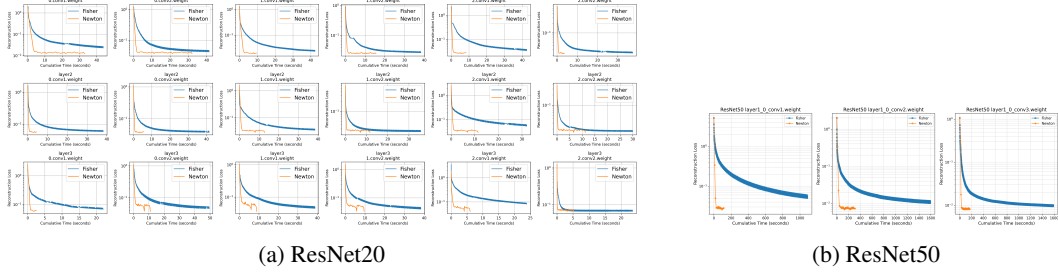

(a) ResNet20

(b) ResNet50

Figure 9: Comparison of Newton's method and Fisher approximation for all layers in ResNet20 and the first three layers of ResNet50.

### A.2.3 EFFECT OF THE NUMBER OF CG ITERATIONS

The plot in Figure 10 shows the impact of varying the maximum number of CG iterations in Algorithm 1 on the reconstruction loss across different layers and models. As expected, decreasing

the maximum CG iterations generally results in slower progress per iteration, as lower caps (e.g., CG 5) reduce the loss at a slower rate compared to higher iteration limits (e.g., CG 500). However, despite this slower per-iteration progress, CG with fewer iterations often converges faster overall, as the algorithm compensates by taking a more iterative approach, covering more batches quickly while solving the second order approximation for each batch more approximately. This means that, although each step is smaller, CG with lower iteration caps refines the solution more frequently, leading to comparable or even faster convergence relative to higher iteration caps.

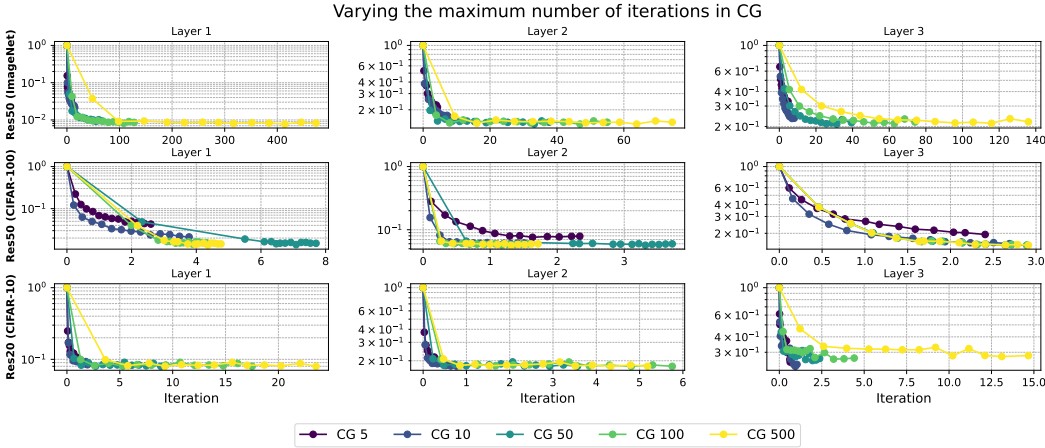

Figure 10: Normalized reconstruction loss starting from the masked dense weights. The plot shows the effect of varying the maximum CG iterations in Algorithm 1. While fewer iterations result in slower progress per step, they often lead to faster overall convergence by refining the solution more frequently.

### A.2.4 Effect of the parameter $K$ on run-time

We provide in Table 3 a comparison of (1) run-time in minutes for pruning the entire network and (2) peak GPU memory usage, as the parameter $K$ varies. In practice $K$ should be chosen as large as possible while not exceeding available GPU memory or run-time expectations. In practice, moderate values of $K$ can lead to good performance while having very small run-times.

| | **ResNet20 (CIFAR-10)** | | **ResNet50 (CIFAR-100)** | | **ResNet50 (ImageNet-1k)** | |
|---|---|---|---|---|---|---|
| $K$ | Time | Peak Memory (GB) | Time | Peak Memory (GB) | Time | Peak Memory (GB) |
| 0 | 0.7 | 2.2 | 11.3 | 8.7 | 10.5 | 8.6 |
| 1 | 0.8 | 2.5 | 10.7 | 10.8 | 12.9 | 15.1 |
| 3 | 1.6 | 2.8 | 11.1 | 11.4 | 25.2 | 17.3 |
| 5 | 2.3 | 3.3 | 11.4 | 12.1 | 39.7 | 25.8 |
| 10 | 4.8 | 3.3 | 13.3 | 15.5 | 94.8 | 38.3 |
| 20 | 8.1 | 4.3 | 16.1 | 21.9 | 184.1 | 54.0 |
| 30 | 11.2 | 4.9 | 17.0 | 24.9 | 289.5 | 66.9 |
| 40 | 11.8 | 5.1 | 18.8 | 28.1 | OOM | OOM |
| 60 | $>$ max $K$ | $>$ max $K$ | 20.3 | 31.6 | OOM | OOM |

Table 3: Effect of varying $K$ on the run-time and peak memory usage of the full SNOWS algorithm. The run-time shown is in minutes.

### A.3 Implementation and Experimental details

### A.3.1 $N$:$M$ Sparsity in CNNs

$N$:$M$ sparsity enforces a fine-grained sparsity pattern along specific dimensions of the weight tensor. In the context of convolutional neural networks, we apply $N$:$M$ sparsity along the input dimension

$d_{\text{in}}$ while keeping the other dimensions fixed. This means that, for each output channel $n$ and each spatial location in the kernel $(i, j)$, we group the weights $\boldsymbol{W}^\ell_{n,G,i,j}$ along the input channels into groups $G$ of $M$ consecutive weights such that $\left\|\boldsymbol{W}^\ell_{n,G,i,j}\right\|_0 = N$. Within each group, exactly $N$ weights are non-zero, and the remaining $M - N$ weights are zeroed out. An example for 2:4 sparsity is shown in Figure 11.

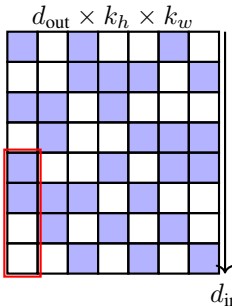

$$d_{\text{out}} \times k_h \times k_w$$

$$d_{\text{in}}$$

Figure 11: An example of a 2:4 sparse convolutional kernel.

### A.3.2 $K$-STEP LAYER-WISE RECONSTRUCTION PROBLEM IN CNNS.

Each convolutional layer typically involves several linear and non-linear operations. The exact operations can differ across architectures, but we present an example for the ResNet architecture (He et al., 2015). The functions in Eqn (3), which captures the operations at layer $\ell$, can be mathematically defined as $f^\ell(\boldsymbol{X}^\ell, \boldsymbol{W}^\ell) = \text{Conv}(\boldsymbol{X}^\ell, \boldsymbol{W}^\ell)$ refers to the convolution operation. The Conv operation can be written as a linear operation so for $K = 0$ this problem reduces to the layer-wise reconstruction problem in Equation 3. For larger $K$, say $K = 1$, we define :

$$f^{\ell:\ell+1}(\boldsymbol{X}^\ell, \boldsymbol{W}^\ell) = \text{ReLU}\left(\text{BN}\left(f^\ell(\boldsymbol{X}^\ell, \boldsymbol{W}^\ell)\right)\right) + \boldsymbol{X}^\ell$$

where BN refers to batch normalization, $\text{ReLU}(\boldsymbol{M}) = \max(\boldsymbol{M}, \boldsymbol{0})$ and the addition of $\boldsymbol{X}^\ell$ is the residual connection. For $K = 1$, we directly jump to reconstruct the output of the ReLU since BN is a small linear rescaling applied to the convolution, so we do not directly reconstruct its output and likewise for residual connections, though these are inherently absorbed in the reconstruction at the output of the ReLU.

### A.3.3 RAW NUMBERS FOR UNSTRUCTURED COMPARISONS

For Figure 3, we use the benchmarking numbers from (Benbaki et al., 2023) and (Meng et al., 2024b). However, their dense versions of ResNet20 and MobileNetV1 have slightly different starting accuracy than ours. This the reason we report relative accuracy to the baseline in Figure 3.

| ResNet20 on CIFAR10 | | | | | | |
|---|---|---|---|---|---|---|
| Sparsity | Baseline Accuracy: 91.36% | | | | | Baseline: 92.56% |
| | MP | WF | CBS | CHITA | FALCON | SNOWS |
| 0.5 | 88.44 (-2.92) | 90.23 (-1.13) | 90.58 (-0.78) | 90.60 (-0.76) | 90.87 (-0.49) | 92.28 (-0.28) |
| 0.6 | 85.24 (-6.12) | 87.96 (-3.40) | 88.88 (-2.48) | 89.22 (-2.14) | 89.67 (-1.69) | 91.90 (-0.66) |
| 0.7 | 78.79 (-12.57) | 81.05 (-10.31) | 81.84 (-9.52) | 84.12 (-7.24) | 84.42 (-6.94) | 90.87 (-1.69) |
| 0.8 | 54.01 (-37.35) | 62.63 (-28.73) | 51.28 (-40.08) | 57.90 (-33.46) | 65.17 (-26.19) | 87.82 (-4.74) |
| 0.9 | 11.79 (-79.57) | 11.49 (-79.87) | 13.68 (-77.68) | 15.60 (-75.76) | 19.14 (-72.22) | 64.80 (-27.76) |
| MobileNetV1 on ImageNet-1K | | | | | | |
| Sparsity | Baseline Accuracy: 71.95% | | | | | Baseline: 70.89% |
| | MP | WF | CBS | CHITA | FALCON | SNOWS |
| 0.5 | 62.61 (-9.34) | 68.91 (-3.04) | 70.21 (-1.74) | 70.42 (-1.53) | 70.35 (-1.60) | 70.10 (-0.79) |
| 0.6 | 41.94 (-30.01) | 60.90 (-11.05) | 66.37 (-5.58) | 67.30 (-4.65) | 67.18 (-4.77) | 68.70 (-2.19) |
| 0.7 | 6.78 (-65.17) | 29.36 (-42.59) | 55.11 (-16.84) | 59.40 (-12.55) | 58.40 (-13.55) | 65.28 (-5.61) |
| 0.8 | 0.11 (-71.84) | 0.24 (-71.71) | 16.38 (-55.57) | 29.78 (-42.17) | 25.82 (-46.13) | 54.83 (-16.06) |

Table 4: Raw accuracy (%) with absolute accuracy change (in brackets) compared to the baseline. The baseline accuracies are shown above the methods.

### A.3.4 RETRAINING EXPERIMENTS

In Table 5, we provide retraining experiments for SNOWS and the best competing method from our unstructured sparsity experiments, FALCON Meng et al. (2024b). The retraining experiments use a step decay learning rate schedule to fine-tune. For CIFAR-10, we retrained ResNet20 for 90 epochs, using SGD with momentum 0.9 and weight decay 5e-5. The learning rate started at 3e-3 and was reduced by a factor of 0.1 every 30 epochs. For ImageNet-1K, we retrained MobileNetV1 for 5 epochs using the same optimizer settings, with the learning rate decaying every 2 epochs. Throughout retraining, we maintained the pruning mask to preserve the network sparsity, allowing only non-zero weights to be updated. As shown in the results table, this retraining strategy was particularly effective at high sparsity levels (0.8-0.9), where SNOWS significantly outperformed FALCON in retaining model accuracy.

| Dataset | Sparsity | FALCON | | SNOWS | |
|---|---|---|---|---|---|
| | | Original | +RT | Original | +RT |
| ResNet20 (CIFAR-10) | 0.5 | 90.87 (-0.49) | 91.07 (-0.29) | 92.28 (-0.28) | 92.32 (-0.24) |
| | 0.6 | 89.67 (-1.69) | 90.58 (-0.78) | 91.90 (-0.66) | 92.13 (-0.43) |
| | 0.7 | 84.42 (-6.94) | 89.64 (-1.72) | 90.87 (-1.69) | 91.85 (-0.71) |
| | 0.8 | 65.17 (-26.19) | 87.59 (-3.77) | 87.82 (-4.74) | 91.06 (-1.50) |
| | 0.9 | 19.14 (-72.22) | 81.60 (-9.76) | 64.80 (-27.76) | 87.69 (-4.87) |
| MobileNetV1 (ImageNet-1K) | 0.5 | 70.35 (-1.60) | 70.66 (-1.29) | 70.10 (-0.79) | 70.66 (-0.23) |
| | 0.6 | 67.18 (-4.77) | 68.89 (-3.06) | 68.70 (-2.19) | 70.10 (-0.79) |
| | 0.7 | 58.40 (-13.55) | 64.74 (-7.21) | 65.28 (-5.61) | 68.76 (-2.13) |
| | 0.8 | 25.82 (-46.13) | 53.84 (-18.11) | 54.83 (-16.06) | 64.70 (-6.19) |

Table 5: Raw accuracy (%) and absolute accuracy change (in brackets) for FALCON and SNOWS before and after retraining at different sparsity levels. The baseline accuracies are as given in Table 4.

### A.3.5 INFERENCE TIME SPEED-UPS FOR SEMI-STRUCTURED SPARSITY

Calculating exact inference time speed-ups for N:M sparsity patterns presents significant challenges due to limited native PyTorch support and the need for specialized CUDA kernels to realize speedups. Moreover, currently only the 2:4 pattern is supported by NVIDIA's sparse tensor cores. To

provide meaningful performance metrics, we instead report end-to-end FLOPs reduction across various architectures and sparsity patterns in Table 6, accounting for all network operations and layers.

Table 6: FLOPs and Accuracy Reduction Across Models and Semi-Structuerd Sparsity Patterns

| Model | Dataset | Sparsity Pattern | FLOPs Reduction | Accuracy Change |
|---|---|---|---|---|
| ViT-B-16 | ImageNet-1k | 2:4 | 1.97× | -3.85% |
| ResNet50 | ImageNet-1k | 1:4 | 2.99× | -9.38% |
|  |  | 2:4 | 1.79× | -1.74% |
| ResNet50 | CIFAR-100 | 1:4 | 3.13× | -1.35% |
|  |  | 2:4 | 1.83× | -0.16% |
| ResNet20 | CIFAR-10 | 1:4 | 3.80× | -3.49% |
|  |  | 2:4 | 1.97× | -0.64% |

### A.3.6 HYPERPARAMETER CONFIGURATIONS FOR SNOWS USED IN EXPERIMENTS

| Architecture | Dataset | $K$-Step Value | | Dampening $\lambda$ | CG Tol |
|---|---|---|---|---|---|
|  |  | $N{:}M$ Sparsity | Unstructured Sparsity |  |  |
| ResNet20 | CIFAR-10 | $K = 40$ | $K = 40$ | $1e{-}4$ | $1e{-}3$ |
| ResNet50 | CIFAR-100 | $K = 100$ | — | $1e{-}4$ | $1e{-}3$ |
| ResNet50 | ImageNet-1k | $K = 30$ | — | $1e{-}4$ | $1e{-}3$ |
| MobileNet | ImageNet-1k | — | $K = 60$ | $1e{-}4$ | $1e{-}3$ |
| ViT/B-16 | ImageNet-1k | $K = 3$ | — | $1e{-}4$ | $5e{-}4$ |
| ViT/L-16 | ImageNet-1k | $K = 4$ (QKV) $K = 1$ (MLP) | — | $1e{-}4$ | $5e{-}4$ |

Table 7: $K$-step values, dampening factors, and CG tolerances used for $N{:}M$ sparsity and unstructured sparsity experiments across different architectures.

### A.3.7 SNOWS ALGORITHM RUNTIME

For moderate values of $K$, our approach performs better and can run much faster than prior approaches, and scales much better in terms of architecture size. In Table 6, we compare the runtime of our algorithm to CHITA (Benbaki et al. (2023)). We remark that other competitors WoodFisher and CBS require hours to prune even MobileNet with a small sample size, see section 4.1.1 of (Benbaki et al. (2023)). We show below the run-time for pruning three networks of different sizes: ResNet20 (250k parameters), MobileNet (4m parameters), and ResNet50 (25m parameters). The main advantage of our method in terms of efficiency is scaling to large networks since we avoid computing, inverting, or storing the full Hessian or its approximation, and moreover not even storing or inverting the smaller layer-wise Hessian. Notably, our method outperforms CHITA even with moderate values of $K$ (though larger values used in our final experiments can obtain the best results e.g. in Figure 3).

| Model-Sparsity | Samples $n$ | SNOWS | | CHITA | |
|---|---|---|---|---|---|
| | | Acc (%) | Runtime (min) | Acc (%) | Runtime (min) |
| ResNet50 (25m parameters) | 500 | 75.7 | 9.66 | 73.6 | 231.20 |
| MobileNet (4m parameters) | 1000 | 61.1 | 5.56 | 59.4 | 16.82 |
| ResNet20 (250k parameters) | 1000 | 90.0 | 0.56 | 84.1 | 2.23 |

Table 8: Comparison of performance versus run-time comparing SNOWS and CHITA pruning different architectures to 70% unstructured sparsity. ResNet-20 is run on CIFAR-10, MobileNet on ImageNet-1k, and ResNet50 on CIFAR-100. We use 1000 samples for ResNet20 and MobileNet, but for ResNet50 we use 500 samples since this was the only sample size below which CHITA runs in <5 hours.

### A.3.8 LAYER-WISE RECONSTRUCTION PROBLEMS FOR VIT MODULES.

**MHSA Pruning.** In a Transformer model with $L$ layers, the input to layer $\ell$ is denoted as $\boldsymbol{X}^\ell \in \mathbb{R}^{n \times d}$, where $n$ is the sequence length and $d$ is the embedding dimension. The Multi-Head Self-Attention (MHSA) mechanism at layer $\ell$ uses multiple attention heads to capture different aspects of the input sequence. Specifically, the input $\boldsymbol{X}^\ell$ is linearly projected to produce the concatenated queries ($\boldsymbol{Q}^\ell$), keys ($\boldsymbol{K}^\ell$), and values ($\boldsymbol{V}^\ell$) across all heads:

$$\boldsymbol{Q}^\ell = \boldsymbol{X}^\ell \boldsymbol{W}_Q^\ell, \quad \boldsymbol{K}^\ell = \boldsymbol{X}^\ell \boldsymbol{W}_K^\ell, \quad \boldsymbol{V}^\ell = \boldsymbol{X}^\ell \boldsymbol{W}_V^\ell$$

where $\boldsymbol{W}_Q^\ell, \boldsymbol{W}_K^\ell, \boldsymbol{W}_V^\ell \in \mathbb{R}^{d \times H d_H}$ are the learnable weight matrices for all heads, with $H$ being the number of heads and $d_H$ the dimensionality of each head. The self-attention mechanism then computes the relevance of each token in the sequence by performing the scaled dot product of queries and keys, followed by a softmax operation to produce normalized attention scores:

$$\boldsymbol{Y}^\ell = \text{softmax}\left(\frac{\boldsymbol{Q}^\ell \boldsymbol{K}^{\ell\top}}{\sqrt{d_k}}\right) \boldsymbol{V}^\ell$$

Here, $\boldsymbol{Y}^\ell$ represents the output of the MHSA layer at the current layer $\ell$. In the general multi-step layer-wise reconstruction problem for $K \geq 0$, we aim to reconstruct the outputs over the subsequent $K$ layers while optimizing jointly over the query, key and value matrices. This is expressed as:

$$\min_{\widehat{\boldsymbol{W}}^\ell \in \mathcal{W}} \sum_{k=0}^{K} \left\| \boldsymbol{Y}^{\ell+k} - f^{\ell:\ell+k}(\boldsymbol{X}^\ell, (\widehat{\boldsymbol{W}}_Q^\ell, \widehat{\boldsymbol{W}}_K^\ell, \widehat{\boldsymbol{W}}_V^\ell)) \right\|_2^2$$

where $f^{\ell:\ell+k}(\boldsymbol{X}^\ell, (\widehat{\boldsymbol{W}}_Q^\ell, \widehat{\boldsymbol{W}}_K^\ell, \widehat{\boldsymbol{W}}_V^\ell))$ represents the transformation of the input $\boldsymbol{X}^\ell$ through $K$ layers which can include the FFN outputs, GeLU activations, Layer Normalization outputs, and subsequent attention operations. For a specific case: when $K = 0$, we focus on reconstructing only the output of the attention block in layer $\ell$, which can be written as:

$$\min_{\widehat{\boldsymbol{W}}^\ell \in \mathcal{W}} \left\| \boldsymbol{Y}^\ell - \text{softmax}\left(\frac{\boldsymbol{X}^\ell \widehat{\boldsymbol{W}}_Q^\ell (\boldsymbol{X}^\ell \widehat{\boldsymbol{W}}_K^\ell)^\top}{\sqrt{d_k}}\right) \boldsymbol{X}^\ell \widehat{\boldsymbol{W}}_V^\ell \right\|_2^2$$

Through joint optimization we inherently account for the interactions between the pruned weights. Moreover, since we solve the problem across all attention heads, we can learn redundancies that exist across multiple heads, thereby preserving the overall expressiveness of the MHSA block after pruning.

**Output Projection Pruning.** In MHSA, after the attention operation, the concatenated outputs of the attention heads are linearly projected back to the original embedding dimension $d$ via an output projection matrix $\boldsymbol{W}_O^\ell \in \mathbb{R}^{H d_H \times d}$. This matrix reduces the dimensionality of the concatenated outputs $\boldsymbol{Z}^\ell = \text{concat}(\boldsymbol{Z}_1^\ell, \ldots, \boldsymbol{Z}_H^\ell)$, where each $\boldsymbol{Z}_h^\ell$ is the output from one of the $H$ heads. The output projection is computed as:

$$\boldsymbol{Y}^\ell = \boldsymbol{Z}^\ell \boldsymbol{W}_O^\ell$$

In the case where $K = 0$, the output projection pruning reduces to a standard linear least squares reconstruction problem.

$$\min_{\widehat{\boldsymbol{W}}_O^\ell \in \mathcal{W}} \left\| \boldsymbol{Y}^\ell - \boldsymbol{Z}^\ell \widehat{\boldsymbol{W}}_O^\ell \right\|_2^2$$

For larger $K$, the output projection reconstruction problem involves reconstructing the outputs of the MLP, Attention blocks that proceed it.

**FFN Pruning**. In vision transformers, feedforward networks (FFNs) are responsible for a substantial portion of the overall model parameters, often accounting for more than half of the total parameter count and FLOPS (Kao et al., 2022). The FFN block applies an initial linear transformation, followed by a GeLU activation function (Hendrycks & Gimpel, 2023), followed by an additional linear transformation. To optimize and prune the FFN, we break the MLP into its constituent linear parts, addressing each separately. Specifically, we handle the first linear layer (MLP0) and the second linear layer (MLP3) as distinct optimization tasks. As special cases, for $K = 0$, both MLP0 and MLP3 reduce to a standard linear reconstruction problem:

$$\min_{\widehat{\boldsymbol{W}}^\ell \in \mathcal{W}} \left\| \boldsymbol{X}^\ell \boldsymbol{W}^\ell - \boldsymbol{X}^\ell \widehat{\boldsymbol{W}}^\ell \right\|_2^2$$

For $K = 1$, MLP0 becomes a non-linear problem involving the GeLU activation function:

$$\min_{\widehat{\boldsymbol{W}}^\ell \in \mathcal{W}} \left\| \text{GeLU}(\boldsymbol{X}^\ell \boldsymbol{W}^\ell + \boldsymbol{b}^\ell) - \text{GeLU}(\boldsymbol{X}^\ell \widehat{\boldsymbol{W}}^\ell + \boldsymbol{b}^\ell) \right\|_2^2$$

For larger $K$, the MLP reconstruction problem involves reconstruction subsequent attention operations and MLP blocks. In all cases, MLP0 and MLP3 are optimized independently.

