# OpenReview forum: "Preserving Deep Representations in One-Shot Pruning: A Hessian-Free Second-Order Optimization Framework"
_ICLR.cc/2025/Conference — ICLR 2025 Poster_

### Official Review · Reviewer_YAxu · 2024-10-27

**Soundness:** 3
**Presentation:** 2
**Contribution:** 2
**Rating:** 6
**Confidence:** 4

**Summary:**

This paper proposes SNOWS, a one-shot pruning framework aimed at reducing vision network inference costs without retraining. Unlike traditional approaches that prioritize layer-wise error minimization, SNOWS optimizes a global objective that captures deeper, nonlinear network representations. To efficiently solve the resulting complex optimization problem, SNOWS leverages a Hessian-free second-order method that computes Newton descent steps without needing the full Hessian matrix. SNOWS also enhances any pre-existing sparse mask by refining weights to better leverage nonlinearities in deep features. The framework demonstrates state-of-the-art results across multiple one-shot pruning benchmarks, including residual networks and Vision Transformers. Results are shown for CIFAR10/100 and Mini-Imagenet.

**Strengths:**

**===After rebuttal, I raised score from 5 to 6=====**
+ A method with a new pruning strategy at the level of global reconstruction shows promising results on common datasets CIFAR and Mini-Imagenet.
+ The proposed framework is reportedly efficient where it does not require storing the full Hessian matrix, allowing it to perform running with large neural networks.
+ Providing more details of pruning with ViT models gives some new insights.

**Weaknesses:**

**Concerns**
+ The baselines seem to be out-of-date, it would be helpful to specify additional recent baselines from 2024. More baselines should be considered for example [1,2].

+ Considering specific aspects of the method that could be condensed or moved to an appendix, and particular experimental analyses or comparisons should be expanded:  Accuracy and latency speed-up with different network architectures: Res50, MobileNet, ViT.

+ Section 5, "Experimental setup" is written too long, it contains also the results and a description of results that is not "setup". It should be separated into a new paragraph or subsection. More breaks between paragraphs should be added for better readability.

+ Considering particular efficiency metrics the authors could report (e.g., runtime, memory usage) and specific methods (in Table 3) they should compare against to demonstrate their efficiency claims.

**Other suggestions**
+ Mentioning that experiments on ImageNet-1k (lines 102, 444)  might be misleading since the fact that the authors only use the Mini-ImageNet with 10,000 samples compared to ImageNet-1k (1.2M samples). Replacing ImageNet-1k with MiniImagenet-1k is more appropriate.

+ There are two "Experimental setup" parts which are the same level as the network part (CNN running, Vision transformers, ...), which might cause confusion, suggesting making it a subsection for each type of network.

+ The so-called "Figure 3" (Line 472-484) should be a "Table 3". The text referred to that table should be revised accordingly.

+ For better readability, I suggest rather "minimize (7)" --> "minimize Eq. 7", Taylor expansion in (7) --> "Taylor expansion in Eq. 7", each call to (5) --> "each call to Eq. 5", etc.

+ Equations are written in a separate line with a center and should be enumerated, for example for line 214, line 382, and 384.

+ Although it is lookable, Figure 4 should make the text inside the figure larger for better readability.

[1] FALCON: FLOP-Aware Combinatorial Optimization for Neural Network Pruning, AISTATS 2024

[2] LayerMerge: Neural Network Depth Compression through Layer Pruning and Merging, ICML 2024

**Questions:**

Suggest that the authors define the MP metric/method when it is first used and provide a citation if it refers to a specific published method.

---

> ### Author Response · Authors · 2024-11-22
>
> We thank the reviewer for the close reading of our paper and the detailed comments. We have made most of the suggestions you proposed especially formatting related and provided an updated pdf. We address your concerns below:
>
> **Out of date baselines**
>
> Below we provide the new baseline (FALCON [1]) and include metrics for retraining with unstructured sparsity on on CIFAR-10 and MiniImageNet-1k. We provide the raw-one shot performance and performance after retraining. We retrain for 90 epochs on CIFAR-10 and 5 epochs on ImageNet-1k.
>
> Note: for the retraining experiments, we retrain on the full ImageNet-1k dataset, as opposed to the sample of 3k samples from MiniImageNet-1k we used in our one-shot pruning experiments.
>
> Also, note that FALCON does not actually prune to the target sparsity level given. It's sparsity is less (more dense) than the amount shown, since it uses a flop-aware formulation. As a result, these results are conservative with respect to our outperformance of FALCON.
>
> ## ResNet20 on CIFAR10
>
> | **Sparsity** | **MP**   | **WF**   | **CBS**  | **CHITA** | **FALCON** | **FALCON+RT** | **SNOWS**  | **SNOWS+RT** |
> |--------------|----------|----------|----------|-----------|------------|---------------|------------|--------------|
> | 50%          | -2.92    | -1.13    | -0.78    | -0.76     | -0.49      | -0.29         | -0.28      | **-0.24**    |
> | 60%          | -6.12    | -3.40    | -2.48    | -2.14     | -1.69      | -0.78         | -0.66      | **-0.43**    |
> | 70%          | -12.57   | -10.31   | -9.52    | -7.24     | -6.94      | -1.72         | -1.69      | **-0.71**    |
> | 80%          | -37.35   | -28.73   | -40.08   | -33.46    | -26.19     | -3.77         | -4.74      | **-1.50**    |
> | 90%          | -79.57   | -79.87   | -77.68   | -75.76    | -72.22     | -9.76         | -27.76     | **-4.87**    |
>
> ## MobileNetV1 on ImageNet-1K
>
> | **Sparsity** | **MP**   | **WF**   | **CBS**  | **CHITA** | **FALCON** | **FALCON+RT** | **SNOWS**  | **SNOWS+RT** |
> |--------------|----------|----------|----------|-----------|------------|---------------|------------|--------------|
> | 50%          | -9.34    | -3.04    | -1.74    | -1.53     | -1.60      | -1.29         | -0.79      | **-0.23**    |
> | 60%          | -30.01   | -11.05   | -5.58    | -4.65     | -4.77      | -3.06         | -2.19      | **-0.79**    |
> | 70%          | -65.17   | -42.59   | -16.84   | -12.55    | -13.55     | -7.21         | -5.61      | **-2.13**    |
> | 80%          | -71.84   | -71.71   | -55.57   | -42.17    | -46.13     | -18.11        | -16.06     | **-6.19**    |
>
> We remark that though LayerMerge [2] is a good pruning technique, it should not be considered a one-shot pruning method, since the method has close to 10\% accuracy following pruning (cf. Figure 3 of their paper), and requires extensive retraining to recover performance (they use 180 epochs of retraining on ImageNet-1k).
>
> [1] Meng, X., Chen, W., Benbaki, R., & Mazumder, R. (2024). FALCON: FLOP-Aware Combinatorial Optimization for Neural Network Pruning. In Proceedings of The 27th International Conference on Artificial Intelligence and Statistics (AISTATS).
>
> [2] LayerMerge: Neural Network Depth Compression through Layer Pruning and Merging, International Conference on Machine Learning, 2024.

---

> > ### Author Response · Authors · 2024-11-22
> >
> > **Run-time of our method**
> >
> > For the run-time of our method compared to competitors please see our response to Reviewer 96mw ("Run-time of our method").
> >
> > **Inference Speed-ups**
> >
> > It is difficult for us to calculate the exact inference time speed-up for $N$:$M$ sparsity. This is because currently the support for $N$:$M$ sparsity natively in PyTorch is limited, and can require developing specialized CUDA kernels. Moreover, only the 2:4 pattern is supported at present by NVIDIA's sparse tensor cores. However, below we provide the end-to-end FLOP reduction for several of the models and $N$:$M$ patterns we use. Note that this takes in to account all operations and layers in the network, so should give a reasonable estimate of the end-to-end speed-up one could achieve with a specialized kernel. We show below that some architectures are more amenable to speed-ups than others. For example ResNet50 has downsample layers which limit speed-up achievable with only the regular convolutional layers. VIT-B-16 and ResNet20 have no such layers so attain close to the maximum possible speed-up.
> >
> > | **Model**     | **Dataset**     | **Sparsity Pattern** | **FLOPs Reduction** | **Accuracy Change** |
> > |---------------|-----------------|-----------------------|---------------------|---------------------|
> > | ViT-B-16      | ImageNet-1k     | 2:4                   | 1.97×              | -3.85%              |
> > | ResNet50      | ImageNet-1k     | 1:4                   | 2.99×              | -9.38%              |
> > |               |                 | 2:4                   | 1.79×              | -1.74%              |
> > | ResNet50      | CIFAR-100       | 1:4                   | 3.13×              | -1.35%              |
> > |               |                 | 2:4                   | 1.83×              | -0.16%              |
> > | ResNet20      | CIFAR-10        | 1:4                   | 3.80×              | -3.49%              |
> > |               |                 | 2:4                   | 1.97×              | -0.64%              |
> >
> > This is an area of active development and we hope that we will see new developments in 2025 surrounding more native software support for semi-structured sparsity.

---

> > > ### Comment · Reviewer_YAxu · 2024-11-25
> > >
> > > Thanks for the rebuttal. It addressed partly my raised concerns, so I am adjusting the score.

---

> > > > ### Author Response · Authors · 2024-11-25
> > > >
> > > > Dear Reviewer YAxu,
> > > >
> > > > Thank you for your thoughtful review and for raising our score. Your suggestions have helped strengthen the paper.
> > > >
> > > > Best regards,
> > > >
> > > > Authors

---

### Official Review · Reviewer_EtDe · 2024-11-02

**Soundness:** 3
**Presentation:** 3
**Contribution:** 3
**Rating:** 6
**Confidence:** 4

**Summary:**

(+) The authors propose optimizing a more global reconstruction objective referred to as SNOW, which accounts for non-linear activations deep in the network to obtain a better proxy for the network loss.

(+) This nonlinear objective leads to a more challenging optimization problem - the authors demonstrate that it can be solved efficiently using Hessian-free optimization.

(+) SNOWS demonstrates its effectiveness with state-of-the-art results on various one-shot pruning benchmarks, including  CNNs and Vision Transformers (ViTs).

**Strengths:**

(+) SNOW can be seamlessly applied to any sparse mask from prior methods, refining weights to leverage the nonlinearities in the deep feature representations of CNNs and ViTs.

(+) The updated rule (Newton) and the illustrative examples in the Supplementary section are well articulated.

**Weaknesses:**

(-) The motivations for the improved layer-wise model (Eq. 3) need clarification. Why are the subsequent operations $f^{l:l+K}$ necessary-are they intended to minimize residuals? Please clarify the motivation behind SNOW by explaining the concept of Fig. 2 and including additional captions.

(-) While SNOW’s performance with increasing K iterations is well documented, computational comparisons with baselines are lacking. For consistent comparisons, all experimental results should explicitly include the number of K-iterations in captions of tables and figures. Moreover, I recommend the authors add an ablation study on the time complexity of K-iterations to the experimental result tables.

**Questions:**

Please see the above weaknesses.

---

> ### Author Response · Authors · 2024-11-22
>
> We thank the reviewer for your positive review and comments. We found your questions to be relevant and useful for improving our work.
>
> **Motivation for the SNOWS loss function**
>
> We thank the reviewer for the question regarding the motivation of including subsequent operations $f^{\ell: \ell+K}$ in the reconstruction loss. Indeed, your intuition is correct that the motivation is in a way to minimize the residuals of the pruned network, relative to the dense network. To give you more intuition, we can return to the original motivation of the layer-wise reconstruction objective given in [1]. In their paper, they show that the error $\tilde{\varepsilon}^L = \frac{1}{\sqrt{n}} || \boldsymbol{Y}^L - \widehat{\boldsymbol{Y}}^L||_F$ of the entire pruned network at the final layer $L$ is bounded as:
>
> $$
> \tilde{\varepsilon}^L \leq \sum_{\ell=1}^{L-1} \left( \prod_{r=\ell+1}^L \left| \left |\widehat{\boldsymbol{W}}^r \right|\right|_F \sqrt{\delta E^\ell} \right) + \sqrt{\delta E^L}
> $$
>
> where $\delta E^\ell$ is the layer-wise error between the outputs at layer $\ell$, i.e., $\delta E^\ell = \frac{1}{n} \left| \left| \boldsymbol{X}^\ell \boldsymbol{W}^\ell - \widehat{\boldsymbol{X}}^\ell \widehat{\boldsymbol{W}}^\ell \right|\right|_F^2$. In other words, the difference between the final output of the dense and pruned network is bounded by a product of the layer-wise errors. This means if the layer-wise errors can be kept small, the overall error should be controlled.
>
> However, there is an interesting subtlety to how the errors propagate. Say we are pruning layer $\ell$. It would seem as though the contribution by layer $\ell$ to $\tilde{\varepsilon}^L$ is confined to $\delta E^\ell$. However, notice that
>
> $$
> \widehat{\boldsymbol{X}}^{\ell+1} = \widehat{\boldsymbol{Y}}^{\ell} = f^{\ell}(\widehat{\boldsymbol{X}}^{\ell}, \widehat{\boldsymbol{W}}^{\ell})
> $$
>
> So pruning at the current layer changes the input at the next layer and moreover at all subsequent layers since $\widehat{\boldsymbol{X}}^{\ell+k} = \widehat{\boldsymbol{Y}}^{\ell+k-1} = f^{\ell+k-1}(\widehat{\boldsymbol{X}}^{\ell+k-1}, \widehat{\boldsymbol{W}}^{\ell+k-1})$, for all $k = 1, \dots, L - \ell$.
>
> So after pruning layer $\ell$, we can observe the expression for the error in the subsequent layers
>
> $$
> \delta E^{\ell+k} = \frac{1}{n} \left| \left| \boldsymbol{X}^{\ell+k} \boldsymbol{W}^{\ell+k} - \widehat{\boldsymbol{X}}^{\ell+k} \widehat{\boldsymbol{W}}^{\ell+k} \right| \right|_F^2
> $$
>
> At the time of pruning layer $\ell$, $\boldsymbol{W}^{\ell+k} = \widehat{\boldsymbol{W}}^{\ell+k}$, since these weights have not yet been pruned, but notice that already $\boldsymbol{X}^{\ell+k} \neq \widehat{\boldsymbol{X}}^{\ell+k}$, since our pruning decision at layer $\ell$ has affected all layers after it.
>
> The error term for the future layers becomes
>
> $$
> \delta E^{\ell+k} = \frac{1}{n} \left| \left|(\boldsymbol{X}^{\ell+k} - \widehat{\boldsymbol{X}}^{\ell+k}) \boldsymbol{W}^{\ell+k} \right|\right|_F^2
> $$
>
> Hence the error at the future layers is determined by the magnitude of the difference $\boldsymbol{X}^{\ell+k} - \widehat{\boldsymbol{X}}^{\ell+k}$ induced by pruning layer $\ell$.
>
> Our formulation, by minimizing
>
> $$\sum_{k=0}^K| | \mathbf{Y}^{\ell+k} -\widehat{\mathbf{Y}}^{\ell+k}||_F^2 $$
>
> $$= \sum_{k=0}^K||\mathbf{X}^{\ell+k+1} -\widehat{\mathbf{X}}^{\ell+k+1}||_F^2$$
>
> can be interpreted as directly minimizing the subsequent error terms $\sum_{k=0}^K \delta E^{\ell+k}$ during the pruning of layer $\ell$ as opposed to only $\delta E^\ell$.
>
> [1] Dong, X., Chen, S., & Pan, S. J. (2017). Learning to Prune Deep Neural Networks via Layer-wise Optimal Brain Surgeon. In Advances in Neural Information Processing Systems

---

> ### Author Response · Authors · 2024-11-22
>
> **The effect of $K$ on run-time**
>
> With regards to the ablations on the parameter $K$, we provide below a comparison of (1) run-time for pruning the entire network and (2) peak GPU memory usage, as the parameter $K$ varies for ResNet20 (R20) on CIFAR-10 and ResNet50 (R50) on ImageNet-1k. We have found this a useful ablation study and will include the comparisons in the final version of the paper. Our guidance is to choose the parameter $K$ to be as large as possible
>
> | $K$ |       **R20 (CIFAR-10)** |            | **R50 (CIFAR-100)** |            | **R50 (ImageNet-1k)** |            |
> |-----|---------------------|------------|---------------------|------------|-----------------------|------------|
> |     | Time (mins)           | Mem (GB)   | Time (mins)           | Mem (GB)   | Time (mins)             | Mem (GB)   |
> |-----|---------------------|------------|---------------------|------------|-----------------------|------------|
> | 0   | 0.7                | 2.2        | 11.3               | 8.7        | 10.5                 | 8.6        |
> | 1   | 0.8                | 2.5        | 10.7               | 10.8       | 12.9                 | 15.1       |
> | 3   | 1.6                | 2.8        | 11.1               | 11.4       | 25.2                 | 17.3       |
> | 5   | 2.3                | 3.3        | 11.4               | 12.1       | 39.7                 | 25.8       |
> | 10  | 4.8                | 3.3        | 13.3               | 15.5       | 94.8                 | 38.3       |
> | 20  | 8.1                | 4.3        | 16.1               | 21.9       | 184.1                | 54.0       |
> | 30  | 11.2               | 4.9        | 17.0               | 24.9       | 289.5                | 66.9       |
> | 40  | 11.8               | 5.1        | 18.8               | 28.1       | OOM                  | OOM        |
> | 60  | >max $K$           | >max $K$   | 20.3               | 31.6       | OOM                  | OOM        |
>
> *Table 1: Memory and run-time using different $K$ in the SNOWS loss function.*
>
> Moreover, we centralize the values of $K$ used in all experiments with other hyperparameters in a single table for ease of comparison:
>
> | **Architecture**   | **Dataset**     | **$K$ (N:M Sparsity)** | **$K$ (Unstructured Sparsity)** | **Dampening ($\lambda$)** | **CG Tol**          |
> |---------------------|-----------------|-----------------------------------|--------------------------------------------|---------------------------|---------------------|
> | ResNet20           | CIFAR-10        | $K =40$                         | $K = 40$                                    | $1\mathrm{e}{-4}$         | $1\mathrm{e}{-3}$   |
> | ResNet50           | CIFAR-100       | $K = 100$                        | ---                                        | $1\mathrm{e}{-4}$         | $1\mathrm{e}{-3}$   |
> | ResNet50           | ImageNet-1k     | $K = 30$                         | ---                                        | $1\mathrm{e}{-4}$         | $1\mathrm{e}{-3}$   |
> | MobileNet          | ImageNet-1k     | ---                              | $K = 60$                                    | $1\mathrm{e}{-4}$         | $1\mathrm{e}{-3}$   |
> | ViT/B-16           | ImageNet-1k     | $K = 3$                          | ---                                        | $1\mathrm{e}{-4}$         | $5\mathrm{e}{-4}$   |
> | ViT/L-16 (QKV)     | ImageNet-1k     | $K = 4$                          | ---                                        | $1\mathrm{e}{-4}$         | $5\mathrm{e}{-4}$   |
> | ViT/L-16 (MLP)     | ImageNet-1k     | $K = 1$                          | ---                                        | $1\mathrm{e}{-4}$         | $5\mathrm{e}{-4}$   |
>
> *Table 2: $K$-step values, dampening factors, and CG tolerances used for $N$:$M$ sparsity and unstructured sparsity experiments across different architectures.*

---

> > ### Comment · Reviewer_EtDe · 2024-11-27
> >
> > Thank you for your explanation of SNOW's motivation and extensive experiments on time and memory complexity. I will keep my score since additional ablation studies regarding time, memory, and performance remain.
> >
> > Best Regards,
> >
> > Reviewer EtDe.

---

> > > ### Author Response · Authors · 2024-11-27
> > >
> > > Dear Reviewer EtDe,
> > >
> > > We thank you again for your feedback on our work. We incorporated several of your suggested changes in the revised manuscript, and we believe it has strengthened the work.
> > >
> > > Best regards,
> > >
> > > Authors

---

### Official Review · Reviewer_96mw · 2024-11-03

**Soundness:** 3
**Presentation:** 3
**Contribution:** 3
**Rating:** 8
**Confidence:** 4

**Summary:**

This paper proposes a novel pruning method called SNOWS (Stochastic Newton Optimal Weight Surgeon), which is an adaptation of pruning methods like the Optimal Brain Surgeon method, that uses the second-order information of the weights to do the pruning.
However, SNOWS does not calculate the full hessian but merely the hessian products, and those too are not all calculated (approximated) at once and the optimization is performed using stochastic newton descent.

**Strengths:**

The following are the strengths of the paper:
1. The idea contributed by this paper seems to be novel and is very interesting.

2. The theoretical aspect of the paper seems sound.

3. The proposed method seems universally applicable to convolution and ViT-based models.

4. The writing of the paper is quite clear.

**Weaknesses:**

The paper's introduction makes some glaring errors in defining concepts. For example, it describes one-shot pruning methods as pruning methods that do not require retraining after pruning. However, this is not accurate. Pruning can be done iteratively or one-shot; however, some finetuning might be required after one-shot pruning, as shown by [1], [2], and many other methods covered by [3].

With the above definition for one-shot in mind, [4] achieves unstructured pruning with more than 90% sparsity while not dropping any clean performance for multiple architectures. The proposed SNOWS method should also be evaluated against [4] for unstructured pruning performance.


Lastly, the major motivation of the proposed work is gain in speed when performing pruning (as the entire hessian need not be calculated), however these no exists latency comparison to other pruning methods.

Improving on these weaknesses might require a major rewrite, and thus the recommendation.

**References**

[1] Sanh, Victor, Thomas Wolf, and Alexander Rush. "Movement pruning: Adaptive sparsity by fine-tuning." Advances in neural information processing systems 33 (2020): 20378-20389.

[2] Sehwag, Vikash, et al. "Hydra: Pruning adversarially robust neural networks." Advances in Neural Information Processing Systems 33 (2020): 19655-19666.

[3] Hoefler, Torsten, et al. "Sparsity in deep learning: Pruning and growth for efficient inference and training in neural networks." Journal of Machine Learning Research 22.241 (2021): 1-124.

[4] Hoffmann, Jasper, et al. "Towards improving robustness of compressed cnns." ICML Workshop on Uncertainty and Robustness in Deep Learning (UDL). Vol. 2. 2021.

**Questions:**

Q1- Since this method is able to achieve semi-structured sparsity, how does its latency compare to unstructured pruning methods? A comprehensive latency study would be very helpful.

---

> ### Author Response · Authors · 2024-11-22
>
> We thank the reviewer for the feedback on our paper. We appreciate your insights and addressed your concerns below:
>
> **Definition of one-shot pruning**
>
> In our introduction, we refer to  *one-shot* pruning as "approaches that attempt to maintain network performance without requiring retraining." Later, in our section on related work, we provide more detail saying "Motivated by the cost of retraining, there has been a rise in methods for the post-training, one-shot pruning problem. In this setting, one typically assumes access to a limited sample of calibration data (e.g. a few thousand examples) and the goal is to compress the network with minimal accuracy loss relative to the dense model."
>
> ***We now cite several works that share similar definitions of one-shot pruning***
>
> Optimal Brain Compression (OBC) [1] write "*An alternative but challenging scenario is the post-training compression setup, in which we are given a trained but uncompressed model, together with a small amount of calibration data, and must produce an accurate compressed model in one shot, i.e., a single compression step, without retraining, and with limited computational costs.*"
>
> SparseGPT [2] define one-shot pruning as: "*compressing the model without retraining"* and write *"Post-training pruning is a practical scenario where we are given a well-optimized model $\theta^{\star}$, together with some calibration data, and must obtain a compressed (e.g., sparse and/or quantized) version of $\theta^{\star}$*"
>
> WoodFisher [3] define "*One-shot pruning, in which the model has to be compressed in a single step, without any
> retraining*"
>
> We remark that these are only a handful of the most cited pruning papers but there are several others with similar definitions e.g. [4, 5, 6, 7].
>
> **The motivation behind one-shot pruning**
>
> We have no doubt that fine-tuning, retraining, or gradual pruning approaches can improve performance, as has been shown by the paper you provided and others e.g. [8], as well as the experiments we provide here in our response. However, this is typically considered a different setting where one has access to the resources required to compute gradients of the full model. The ImageNet dataset is 150GB in size, with over 1.28m images. We are able to accurately prune on this dataset with just 3k samples for ResNet50, and up to 20k for the VIT models, all using a single A100 GPU. Thus one-shot pruning approaches are well-motivated by the need for practitioners to prune without excessive resources. Similarly, a more accurate model requires less retraining and can be retrained to higher performance, as we show in our additional retraining experiments (see our response to reviewer TGxm "Whether retrainining with SNOWS improves performance").
>
> [1] Frantar, E., Singh, S. P., & Alistarh, D. (2022). Optimal Brain Compression: A Framework for Accurate Post-Training Quantization and Pruning. In Advances in Neural Information Processing Systems
>
> [2] Frantar, E., & Alistarh, D. (2023). SparseGPT: Massive Language Models Can Be Accurately Pruned in One-Shot. In Proceedings of the 40th International Conference on Machine Learning
>
> [3] Singh, S. P., & Alistarh, D. (2020). WoodFisher: Efficient Second-Order Approximation for Neural Network Compression. In Advances in Neural Information Processing Systems
>
> [4] Kwon, W., Kim, S., Mahoney, M. W., Hassoun, J., Keutzer, K., & Gholami, A. (2022). A Fast Post-Training Pruning Framework for Transformers. In Advances in Neural Information Processing Systems
>
> [5] Meng, X., Behdin, K., Wang, H., & Mazumder, R. (2024). ALPS: Improved Optimization for Highly Sparse One-Shot Pruning for Large Language Models
>
> [6] Benbaki, R., Chen, W., Meng, X., Hazimeh, H., Ponomareva, N., Zhao, Z., & Mazumder, R. (2023). Fast as CHITA: Neural Network Pruning with Combinatorial Optimization. In Proceedings of the 40th International Conference on Machine Learning
>
> [7] Frantar, E., Kurtic, E., & Alistarh, D. (2021). M-FAC: Efficient Matrix-Free Approximations of Second-Order Information. In Advances in Neural Information Processing Systems
>
> [8] Le, D. H., & Hua, B.-S. (2021). Network Pruning That Matters: A Case Study on Retraining Variants. In Proceedings of the 9th International Conference on Learning Representations

---

> ### Author Response · Authors · 2024-11-22
>
> **The reviewer's claim that [9] achieves 90% sparsity without accuracy loss**
>
> While it is true that [9] achieve 90\% sparsity without dropping any clean performance for multiple architectures, these results are not directly comparable to the baselines provided in our work. This is because [9] uses ImageNet100, a subset of ImageNet with only 100 classes, which makes it easier for large networks to maintain good accuracy even at high sparsity levels after retraining. To verify this, we replicate the dataset from [9] with the exact classes they use and prune and retrain using SNOWS. We utilize the open-access ResNet18 provided by PyTorch, which is the same architecture used in their paper. Starting from a baseline accuracy of 88.36\%, we prune the network to various sparsity levels and retrain for 50 epochs. The results are summarized in the Table below. As shown, even at 90\% sparsity, the network attains an accuracy of 88.96\% (an improvement of +0.6\% over the dense model), and at 95\% sparsity, the accuracy is 84.84\% (a drop of -3.52\%). The primary reason pruning works so effectively here is because the network is very large (11.7m parameters) relative to the number of classes since this is a small subset of ImageNet. In contrast, the networks we and other benchmarks used in our one-shot pruning experiments are generally *closer to capacity* on the datasets we prune on. For example, the ResNet20 we use on CIFAR-10 has 250k parameters and the MobileNet we use for the entire 1000 ImageNet classes has 4m parameters (less than half used for 100 classes in [9]).
>
> | **Sparsity (%)** | **Accuracy (%)** | **Change in Accuracy (%)** |
> |-------------------|------------------|----------------------------|
> | (Dense)          | 88.36           | --                         |
> | 90               | 88.96           | +0.60                     |
> | 93               | 87.18           | -1.18                     |
> | 95               | 84.84           | -3.52                     |
>
> *Table 1: Pruning with SNOWS and Retraining with ResNet18 on ImageNet100, following the setup in [9].*
>
>
> [9] Hoffmann, J., Agnihotri, S., Saikia, T., & Brox, T. (2021). Towards improving robustness of compressed CNNs. In Proceedings of the ICML Workshop on Uncertainty and Robustness in Deep Learning

---

> ### Author Response · Authors · 2024-11-22
>
> **Run-time of our method**
>
> The run-time of our method depends mostly on the value of $K$ taken in the loss function. Below we show the run-time and peak memory consumption as the parameter $K$ varies for ResNet20 (R20) on CIFAR-10 and ResNet50 (R50) on CIFAR100 and ImageNet-1k.
>
> | $K$ |       **R20 (CIFAR-10)** |            | **R50 (CIFAR-100)** |            | **R50 (ImageNet-1k)** |            |
> |-----|---------------------|------------|---------------------|------------|-----------------------|------------|
> |     | Time (min)           | Mem (GB)   | Time (min)           | Mem (GB)   | Time (min)             | Mem (GB)   |
> |-----|---------------------|------------|---------------------|------------|-----------------------|------------|
> | 0   | 0.7                | 2.2        | 11.3               | 8.7        | 10.5                 | 8.6        |
> | 1   | 0.8                | 2.5        | 10.7               | 10.8       | 12.9                 | 15.1       |
> | 3   | 1.6                | 2.8        | 11.1               | 11.4       | 25.2                 | 17.3       |
> | 5   | 2.3                | 3.3        | 11.4               | 12.1       | 39.7                 | 25.8       |
> | 10  | 4.8                | 3.3        | 13.3               | 15.5       | 94.8                 | 38.3       |
> | 20  | 8.1                | 4.3        | 16.1               | 21.9       | 184.1                | 54.0       |
> | 30  | 11.2               | 4.9        | 17.0               | 24.9       | 289.5                | 66.9       |
> | 40  | 11.8               | 5.1        | 18.8               | 28.1       | OOM                  | OOM        |
> | 60  | >max $K$           | >max $K$   | 20.3               | 31.6       | OOM                  | OOM        |
>
> *Table 2: Memory and run-time using different $K$ in the SNOWS loss function.*
>
> Our guidance is to choose the parameter $K$ to be as large as possible for best performance. However, we have found that even small values of $K$ can outperform prior methods, while having faster run-times. Below we compare the runtime of our algorithm to the fastest (and best performing) competitor from our previous experiments, CHITA [6]. We remark that other competitors WoodFisher and CBS require hours to prune even MobileNet with a small sample size (see section 4.1.1 of [6]). We show below the run-time for pruning three networks of different sizes: ResNet20 (250k parameters), MobileNet (4m parameters), and ResNet50 (25m parameters). The main advantage of our method in terms of efficiency is scaling to large networks since we avoid computing, inverting, or storing the full Hessian or its approximation, and moreover not even storing or inverting the smaller layer-wise Hessian. Notably, our method outperforms CHITA even with moderate values of $K$ (though larger values used in our final experiments can obtain the best results). As shown in Table 1, SNOWS has just a slight run-time advantage for small networks but is more than 20 times as fast for ResNet50 which has 25m parameters, while maintaining a clear performance advantage even at small-moderate $K$.
>
> | **Model-Sparsity**       | **Samples $n$** | **Accuracy (SNOWS) (%)** | **Runtime (SNOWS) (min)** | **Accuracy (CHITA) (%)** | **Runtime (CHITA) (min)** |
> |---------------------------|-----------------|--------------------------|---------------------------|--------------------------|---------------------------|
> | ResNet50 (25m parameters) | 500             | 75.7                   | 9.66                     | 73.6                    | 231.20                   |
> | MobileNet (4m parameters) | 1000            | 61.1                  | 5.56                    | 59.4                    | 16.82                    |
> | ResNet20 (250k parameters) | 1000    | 90.0                 | 0.56                     | 84.1                    | 2.23                     |
>
> *Table 3: Run-time comparison of SNOWS and CHITA  across different architectures pruning to 70% sparsity. For SNOWS, we use $K=10, 3, 3$ respectively for the networks shown. ResNet-20 is run on CIFAR-10, MobileNet on ImageNet-1k, and ResNet50 on CIFAR-100.*
>
> Note: we use 1000 samples for ResNet20 and MobileNet, but for ResNet50 we use 500 samples since this was the only sample size below which CHITA runs in $<$5 hours.

---

> > ### Author Response · Authors · 2024-11-22
> >
> > **Latency Study**
> >
> > Finally, for your concern on latency in semi-structured sparsity, please see our response to Reviewer YAxu ("Inference Speed-ups"). Unfortunately, the native software support for $N$:$M$ sparsity is limited currently. However, we provide estimates of the end-to-end speed-up achievable, and are hopeful that more developments will come from NVIDIA and in other software frameworks e.g. Triton in the coming year.

---

> > > ### Comment · Reviewer_96mw · 2024-11-22
> > >
> > > Dear Authors,
> > >
> > > Thank you very much for the responses. I will look at them in detail and reply to them individually soon.
> > > I see that you have replied with multiple references and evaluations, not just to me but to the other reviewers as well, thank you very much for sharing these, it would take me a couple of days to go over all of these.
> > >
> > > However, in the meantime, I do not see a revised version of the submission addressing the concerns of all the reviewers and including the new evaluations. Is that something planned to be done before the end of the discussion phase?
> > >
> > > Best Regards
> > >
> > > Reviewer 96mw

---

> ### Author Response · Authors · 2024-11-22
>
> Dear Reviewer 96mw,
>
> Thank you for taking the time to go through our responses individually. We very much appreciate it.
>
> We wanted to respond as quickly as possible here to give the reviewers a chance to respond before the discussion periods ends. However, we will be finalizing the manuscript including adding new evaluations before the end of the discussion period.
>
> Many thanks again for your time,
>
> Authors

---

> ### Comment · Reviewer_96mw · 2024-11-26
> **Major Concerns remain largely unaddressed.**
>
> Dear Authors,
>
> Thank you very much for your efforts and detailed responses.
>
> I believe there to be some misunderstanding here when saying, "In our introduction, we refer to one-shot pruning as "approaches that attempt to maintain network performance without requiring retraining." Later, in our section on related work, we provide more detail saying "Motivated by the cost of retraining, there has been a rise in methods for the post-training, one-shot pruning problem."
>
> The definition of one-shot pruning provided by me is indeed correct. Pruning can be either iterative, that is done it multiple shots: first prune 10% and re-train, again prune 10% and re-train, and again prune 10% and re-train. and so on, this is done lets say 5 times could give a model that is 0.9*0.9*0.9*0.9*0.9*100 % dense, that is ~59% dense which is 41% sparse compared to the original dense model.
> Or pruning can be done in a single shot, that is to obtain 41% sparsity, the model is pruned 41% in a single go.
> Now, most single-shot pruning methods require some retraining or finetuning after the one shot of pruning.
> However, this finetuning or retraining can turn out to be expensive, especially for large models, therefore your cited SparseGPT, WoodFischer, and some newer pruning methods try to propose one-shot pruning methods that do not require any finetuning. When doing so, this is explicitly mentioned, as done by both the SparseGPT and WoodFischer paper. I believe this has been misunderstood and therefore misrepresented in this submitted paper.
>
> To further quote both the papers, exactly and with context, in WoodFischer, when the paper says "The first is one-shot pruning, in which the model has to be compressed in a single step, without any re-training.", here "without any re-training" is them highlighting their contribution and not them defining one-shot pruning. And by explicitly mentioning that their method does not do re-training, they are acknowledging that one-shot methods by default do "pruning + re-training".
>
> Next, again in the SparseGPT paper they write as the first line of their abstract, "We show for the first time that large-scale generative pretrained transformer (GPT) family models can be pruned to at least 50% sparsity in one-shot, without any retraining, at minimal loss of accuracy." Here, they emphasize on "without any retraining, at minimal loss of accuracy" because this is a notable contribution, since most one-shot pruning methods require retraining, however, their proposed one-shot method does not require re-training and still does not lose performance, making it a contribution so notable that they use it as the first line of their abstract. Moreover, in the SparseGPT paper, in the introduction, they write "By contrast, the only known **one-shot baseline** which easily extends to this scale, **Magnitude Pruning (Hagiwara, 1994; Han et al., 2015)**, preserves accuracy only until 10% sparsity,". Here Haniwara, 1994 is [R1] and Han et al is [R2].
> [R2] even in their abstract define their method as "Our method prunes redundant connections using a three-step method. First, we train the network to learn which connections are important. Next, we prune the unimportant connections. **Finally, we retrain the network to fine tune the weights of the remaining connections.**"
> Also, [R1] in their work, in Section 2.4 Algorithm: Write step 7 as "Find and remove the least important weight." and then write step 8 as "**Retrain the network.**".
>
> In conclusion of point 1, I believe even the works SparseGPT and WoodFischer do not define one-shot pruning in itself to not have any retraining. Rather, they acknowledge that one-shot pruning methods have retraining, and propose their one-shot pruning methods to be unique as they can perform one-shot pruning without having to fine-tune. Therefore the definition of one-shot pruning in this work is wrong and inconsistent with previous literature and must be changed. (Major Concern)

---

> > ### Comment · Reviewer_96mw · 2024-11-26
> > **[contd.] Major Concerns remain largely unaddressed.**
> >
> > Regarding comparisons to Hoffmann et. al., to the best of my understanding from their appendix, they do use MobileNetV2 and MNASNet for their evaluations, these I believe are small models and comparable to use used in this proposed work. However, I do now see that they achieve only 50% pruning while maintaining i.i.d. accuracy, and use data augmentation and knowledge distillation for that, there directly comparing that to SNOWS might not be fair for SNOWS. Maybe finetuning after SNOWS in a similar fashion might be interesting, however, that can easily be future work and is not necessary for this submission. (Minor Concern)
> >
> >
> > Regarding latency gains, thank you very much for sharing these numbers. These partially answer my question. In "response to Reviewer YAxu ("Inference Speed-ups")" I only see gains in FLOPs, and not gains in latency. I do understand the assumption made here, however, I would like to point out that this is merely an assumption that a reduction in FLOPs correlates with a decrease in latency as well. While this assumption holds true for most cases, it does fail in some cases, for example, in [R3], where their proposed method gave significant reductions in FLOPs, but caused an increase in latency due to how CUDA operations are implemented (Very specifically, in their paper one of their proposals is to replace Channel Attention with their proposed "Simplified Channel Attention" as this significantly reduces the FLOPs, however, as seen in (row 3) Table 2 of [R3], this causes a significant gain in latency). Since the claim of SNOWS is a gain in latency, I would suggest including actual latency numbers in the "response to Reviewer YAxu ("Inference Speed-ups")" table. (Major Concern)
> >
> > Currently, only one of my major concerns has been addressed partially. Therefore, I would keep my current score.
> >
> > Best Regards
> >
> > Reviewer 96mw
> >
> >
> >
> > References:
> >
> > [R1] Hagiwara, Masafumi. "A simple and effective method for removal of hidden units and weights." Neurocomputing 6, no. 2 (1994): 207-218.
> >
> > [R2] Han, Song, Jeff Pool, John Tran, and William Dally. "Learning both weights and connections for efficient neural network." Advances in neural information processing systems 28 (2015).
> >
> > [R3] Chen, Liangyu, Xiaojie Chu, Xiangyu Zhang, and Jian Sun. "Simple baselines for image restoration." In European conference on computer vision, pp. 17-33. Cham: Springer Nature Switzerland, 2022.

---

> > > ### Author Response · Authors · 2024-11-26
> > >
> > > Dear Reviewer 96mw,
> > >
> > > Thank you for your detailed comments and for engaging with us. It seems that your comments are on the following points:
> > >
> > >  1.  Semantics (or definition) of one-shot pruning.
> > >  2.  Pruning efficiency gains vs inference efficiency gains with SNOWS.
> > >
> > > **Semantics of one-shot pruning**
> > >
> > > Thank you to the reviewer for engaging with us on the terminology "one-shot" and its usage in prior work.
> > >
> > > Based on your thoughtful comments and our reading of the papers, it appears that there are indeed different uses or interpretations of the term "one-shot".
> > >
> > > We would like to be as clear as possible and not create confusion around the usage of this terminology.
> > > Therefore, to be as clear as possible, we now refer to the setting of our approach as *one-shot pruning without retraining*.
> > >
> > > We also make it clear that  "one-shot pruning with retraining" has been considered in prior work.
> > >
> > > We have revised the paragraph in our introduction as follows.
> > >
> > > "Many pruning approaches require an expensive retraining or fine-tuning phase following weight removal to recover network performance [2, 3]. Especially as networks scale, practitioners looking to prune and cheaply deploy a model often lack the resources to fully retrain it and may not even have access to the model's dataset or optimizer.
> > > While classical one-shot pruning approaches (where weights are removed in a single step rather than iteratively) typically included retraining [4], recent work has explored one-shot pruning without this computationally expensive step [1, 5, 6]."
> > >
> > > Similarly in our related work section, we have changed the title and content to the following:
> > >
> > > "One-shot pruning without retraining. Recent work has demonstrated the possibility of pruning networks in a single step without requiring expensive retraining [1, 5, 6]. In this setting, one typically assumes access to a limited sample of calibration data (e.g. a few thousand examples) and the goal is to compress the network with minimal accuracy loss relative to the dense model."
> > >
> > > We hope this agrees with your understanding. We remark that this positioning is very similar to that of the OBC paper [1] (see the second paragraph of their introduction).

---

> ### Author Response · Authors · 2024-11-26
>
> **Efficiency during pruning vs Efficiency during inference**
>
> We observe that you made these statements:
>
> This round:
>
> “Since the claim of SNOWS is a gain in latency, I would suggest including actual latency numbers in the "response to Reviewer YAxu ("Inference Speed-ups")" table.”
>
> Previous round:
>
> "Lastly, the major motivation of the proposed work is gain in speed when performing pruning (as the entire hessian need not be calculated), however these no exists latency comparison to other pruning methods.“
>
> This makes us think there is perhaps some confusion here around our usage of the term “efficiency” in the paper.
>
> We would like to clarify that the aim of SNOWs is to improve pruning efficiency. This should be contrasted with a statement that SNOWS results in latency speedups or “inference efficiency”. We do not make such a claim in our paper.
>
> We would like to distinguish between two forms of efficiency, the first we claim in our paper and the second we do not:
>
> (i) *Efficiency during pruning*: The main claim of our paper is a gain in pruning efficiency. By that we mean that due to our optimization framework (new K-step loss function and the Hessian-free Newton approach to solve it), we can get higher model performance for the same run-time. Our claim is a more efficient mathematical framework to perform pruning, which provides lower accuracy loss than prior pruning methods. Performance comparisons given in the main text highlight this contribution. Additionally, we show runtime for pruning is faster than the fastest competing algorithm we benchmark against (CHITA) in Table 8 for the same “one-shot without retraining” setup. We also provide ablation studies on the run-time in Table 3.
>
> (ii) *Efficiency during inference*: Once a sparse network is obtained, different sparsity patterns can realize different inference-time speed-ups on target hardware. Unstructured sparsity does not lead to strong speed-ups during inference; hence, N:M sparsity has been proposed in the literature as a pattern that leads to tangible speed-ups [7]. Our algorithm is agnostic to the mask, but we report accuracy for our method on common N:M sparsity patterns (1:4 and 2:4). Speedups available via our approach will be the same speedups from any existing N:M pattern. Following other papers, we report FLOPs as a proxy for true inference-time speed-up, for example see [8] Table 1 and [9] Table 1, though this is not the focus of our work.
>
> We are happy to provide any additional clarification on our contributions.
>
> We thank the reviewer again for the discussion. We have updated the manuscript with the amended introduction and related work section, which we believe has strengthened the paper.
>
>
> Best regards,
>
> Authors
>
>
> [1] Frantar, E., Singh, S. P., & Alistarh, D. (2022). Optimal Brain Compression: A Framework for Accurate Post-Training Quantization and Pruning. In Advances in Neural Information Processing Systems (NeurIPS).
>
> [2] Liu, Z., Sun, M., Zhou, T., Huang, G., & Darrell, T. (2019). Rethinking the Value of Network Pruning. In International Conference on Learning Representations (ICLR).
>
> [3] Frankle, J., & Carbin, M. (2019). The Lottery Ticket Hypothesis: Finding Sparse, Trainable Neural Networks. In International Conference on Learning Representations (ICLR).
>
> [4] Hagiwara, Masafumi (1994). "A simple and effective method for removal of hidden units and weights." Neurocomputing 6, no. 2: 207-218.
>
> [5] Frantar, E., & Alistarh, D. (2023). SparseGPT: Massive Language Models Can Be Accurately Pruned in One-Shot. In Proceedings of the 40th International Conference on Machine Learning (ICML).
>
> [6] Singh, S. P., & Alistarh, D. (2020). WoodFisher: Efficient Second-Order Approximation for Neural Network Compression. In Advances in Neural Information Processing Systems (NeurIPS).
>
> [7] Sun, Y., Guo, H., Ning, S., & Yu, X. (2022). Accelerating Sparse Deep Neural Network Inference Using GPU Tensor Cores. Proceedings of the 2022 IEEE High Performance Extreme Computing Conference (HPEC).
>
> [8] Zhang, Y., Lin, M., Lin, Z., Luo, Y., Li, K., Chao, F., Wu, Y., & Ji, R. (2023). Learning Best Combination for Efficient N:M Sparsity. Proceedings of the International Conference on Machine Learning.
>
> [9] Zhou, A., Ma, Y., Zhu, J., Liu, J., Zhang, Z., Yuan, K., Sun, W., & Li, H. (2021). Learning N:M Fine-Grained Structured Sparse Neural Networks from Scratch. Proceedings of the International Conference on Learning Representations (ICLR).

---

> > ### Comment · Reviewer_96mw · 2024-11-27
> >
> > Dear Authors,
> >
> > The changes look good and thank you for the clarification.
> >
> > I have now updated my score.
> >
> > Best Regards
> >
> > Reviewer 96mw

---

> > > ### Author Response · Authors · 2024-11-27
> > >
> > > Dear Reviewer 96mw,
> > >
> > > Thank you for your investment in the rebuttal phase. Your feedback no doubt improved the paper. We enjoyed the discussion and sincerely appreciate your willingness to raise your score during the rebuttal.
> > >
> > > Best regards,
> > >
> > > Authors

---

### Official Review · Reviewer_XYMY · 2024-11-04

**Soundness:** 3
**Presentation:** 2
**Contribution:** 2
**Rating:** 5
**Confidence:** 3

**Summary:**

The following paper presents SNOWS, a one-shot post-training pruning technique that performs pruning at a layer-wise level while globally optimizing joint layers within a deep neural network to include nonlinear activation functions within. While it resulted in a more challenging optimization problem due to nonlinearities, it is possible to use the Hessian-free approach integrated with a customized Conjugate Gradient (CG) method that efficiently solves second-order approximation during gradient computation while simultaneously allowing reduced memory requirements. Experiments on various Computer Vision tasks under both CNN-based architecture and Vision transformer highlighted the effectiveness of SNOWS.

**Strengths:**

1. Besides achieving performance improvement compared to other one-shot pruning methods, the proposed method also improves the classification performance in image classification tasks when built on top of existing sparse mask selection algorithms for N:M pruning, denoting its practicability for accelerated training.
2. Highly appreciate the fact that the source code is available for inspection.

**Weaknesses:**

1. There are no conclusions and limitations section in the paper, and the writing in the methods section can be more concise in Section 4, which mainly focuses on explaining the SNOW method. This could leave more space to explain the experiment section in the main paper properly along with the results. Overall, the organization of this paper needs a lot of improvement and it feels like the paper is not ready for publication in the current form.
2. Typos:
- "Like retraining approaches" -> "Like retraining, approaches" (Section 1)
- "network parammeters $\mathbf{W} = (\mathbf{W}^1, ..., \mathbf{W}^1)$ should have been $\mathbf{W} = (\mathbf{W}^1, ..., \mathbf{W}^L)$ (Section 3)

**Questions:**

See the weakness section.

===EDIT: Rating is raised from 3 to 5=====

---

> ### Author Response · Authors · 2024-11-22
>
> We thank the reviewer for reading our paper and in particular for noticing two typos. We have amended these typos in the main text.
>
> **Request for clarification on experiments**
>
> The reviewers strengths of the paper are that "The experiment setup is well-explained and shown in a detailed manner." while their first weakness includes "leave more space to explain the experiment section in the main paper." We would appreciate if the reviewer could provide more detail on exactly what remains to be explained in the experiment section.
>
> **Limitations and conclusions**
>
> While we were limited by space in the main paper, we have decided to add a limitations and conclusions section in the supplementary section. We thank the reviewer for this suggestion.
>
> *This work proposes a new layer-wise pruning framework, acting as a bridge between the layer-wise and global pruning loss functions introduced in prior work. The objective of our modified loss function induces a harder non-linear optimization problem compared to the standard layer-wise formulation. We propose an efficient Hessian-free method for optimizing this objective, and demonstrate its efficiency on large-scale networks. There are several limitations to the proposed work: firstly, SNOWS does not do mask selection. This makes it flexible since it can integrate with existing techniques for mask selection, but we believe using discrete optimization (as is done in ([1], [2]) on the SNOWS loss function may lead to stronger results. Moreover, the method has little dependence on the computational modules involved in the reconstruction loss. As such, the framework could readily be extended to Text-to-Speech or Large Language Models. However, this would require further work to specialize the formulation of the $K$-step problem as we did for e.g. VIT modules.*
>
> [1] Meng, X., Chen, W., Benbaki, R., & Mazumder, R. (2024). FALCON: FLOP-Aware Combinatorial Optimization for Neural Network Pruning. In Proceedings of The 27th International Conference on Artificial Intelligence and Statistics (AISTATS).
>
> [2] Benbaki, R., Chen, W., Meng, X., Hazimeh, H., Ponomareva, N., Zhao, Z., & Mazumder, R. (2023). Fast as CHITA: Neural Network Pruning with Combinatorial Optimization. In Proceedings of the 40th International Conference on Machine Learning.

---

> > ### Comment · Reviewer_XYMY · 2024-11-23
> >
> > Dear Authors,
> >
> > I would like to thank you all for providing a justification and response to my review. It's nice to see that a dedicated section on limitations & conclusions is available now. I also appreciate that typos are fixed.
> >
> > I apologize for not being very detailed with my explanations regarding the first weakness in experimental setups. What I primarily refer to as a lack of explanation in the experiment part is what exactly is pointed out by Reviewer EtDe, related to the details and computational complexity of SNOWS compared with other methods. It will take some time for me to go through the revisions and decide whether it is worthy of publication at the current stage.
> >
> > Best regards,
> >
> >  Reviewer XYMY

---

> > > ### Author Response · Authors · 2024-11-23
> > >
> > > Dear Reviewer XYMY,
> > >
> > > Thank you for your follow-up comments. We sincerely appreciate you elaborating your concerns and taking time to go through the revisions. We hope that shortening Section 4 and adding the limitations & conclusions section has addressed your concern regarding the structure of the paper.
> > >
> > > With respect to computational complexity, we hope the details we added clarify the concerns you share with Reviewer EtDe. Increasing $K$ improves performance (we discuss the reasons why in Section A.1.2, and show this empirically in Figure 1), but increases run-time and memory consumption (we show this in Section A.2.4). Overall, very solid trade-offs between run-time and performance can be obtained under our framework especially for large networks (we show this in Section A.3.7). We believe the framework would be of significant interest to the community given it interpolates nicely between the layer-wise framework (which has become popular due to its efficiency) and the global approaches, which are more accurate but significantly harder to scale.
> > >
> > > We thank you again for your time and consideration.
> > >
> > > Best regards,
> > >
> > > Authors

---

> > > > ### Comment · Reviewer_XYMY · 2024-11-27
> > > >
> > > > Dear Authors,
> > > >
> > > > Thank you for the clarification regarding the experiment setups.
> > > >
> > > > I've checked the most recent version of the paper, and I guess the explanations regarding the experiment section are much more extensive than those of the previous iteration.
> > > >
> > > > I would like to adjust my score based on the current version of the paper.

---

> > > > > ### Author Response · Authors · 2024-11-27
> > > > >
> > > > > Dear Reviewer XYMY,
> > > > >
> > > > > Thank you again for your feedback, particularly regarding the structure of the paper.
> > > > >
> > > > > We appreciate you raising your score. However, we would kindly ask you to outline your remaining concerns, since your current score still reflects a rejection of the work. Otherwise, if you are satisfied with the changes, we ask that you raise your score to reflect this.
> > > > >
> > > > > Best regards,
> > > > >
> > > > > Authors

---

### Official Review · Reviewer_TGxm · 2024-11-04

**Soundness:** 3
**Presentation:** 3
**Contribution:** 3
**Rating:** 8
**Confidence:** 4

**Summary:**

The paper argues it is important to consider the effect of pruning of the whole network instead of just layerwise when performing one-shot post-training pruning. They use a Hessian-free second-order method to optimize their formulation. The results show significant improvement in image-classification tasks on resnets and vits.

**Strengths:**

1. The formulation is neat, and the use of tools from optimization literature (Hessian-free conjugate gradients) is elegant.
2. The results show significant improvements over OBC and pruning using the layerwise objective.
3. Writing is clear and the main claims are supported by the experiments.

**Weaknesses:**

1. The emphasis on using true Hessian information is not supported in the experiments. This is not a crucial point but in most scenarios in deep learning, Fisher matrix-based approximation of Hessian [a] (or even Ada-grad [b] or Adam) works well. It would be worth providing that baseline to better support the proposed method.

- [a] Martens, James. "New insights and perspectives on the natural gradient method." Journal of Machine Learning Research 21.146 (2020): 1-76.
- [b] Duchi, John, Elad Hazan, and Yoram Singer. "Adaptive subgradient methods for online learning and stochastic optimization." Journal of machine learning research 12.7 (2011).

**Questions:**

1. There are a few cases where other pruning methods fail (very low accuracy, Fig. 3, Tab. 1), could you comment on the potential reasons?
2. Does retraining improves the performance further after pruning using the proposed method?

---

> ### Author Response · Authors · 2024-11-22
>
> We sincerely thank the reviewer for their positive and constructive feedback on our work. We address your questions and comments below.
>
> **True Hessian Information versus Fisher approximation**
>
> We ran some additional comparisons, replacing the true Hessian with the Fisher approximation, and comparing the objective value (Eqn 3 in our paper).  Indeed, we do observe a performance improvement for using the true Hessian over the Fisher approximation. The main improvement in using the true Hessian via the Hessian-free approach is speed of convergence. Below we plot the objective value when using the Fisher approximation versus the exact Hessian-free steps for all layers in ResNet20 and the first 3 layers in ResNet50. We find that using Newton step as opposed to the Fisher approximation leads to much faster convergence (not just in iterations but in overall run-time). For the larger ResNet50 networks, running the Fisher approximation to convergence is actually prohibitive.
>
> The figures can be viewed here:
>
> ResNet20: https://github.com/anon-iclr-snows/SNOWS/blob/main/Rebutal-Figures/loss_plot_cifar10.png
>
> ResNet50: https://github.com/anon-iclr-snows/SNOWS/blob/main/Rebutal-Figures/loss_plots_resnet50_layers.png
>
> **Why competing methods fail in some scenarios**
>
> This is generally due to either 1) very high sparsity and/or 2) a restrictive structured sparsity pattern. For example, in Figure 3 in the paper, the competing methods generally fail at 1:4 sparsity. This is a sparsity pattern in which at most 1 of every 4 weights in every contiguous block of 4 are retained. This induces both high sparsity and a restrictive pattern on the non-zeros, which leads to failure without weight readjustment. Intuitively, weight adjustments become more crucial as the sparsity level increases. In Table 1, VITs experience the same effect but fail at even lower sparsity than ResNet models.
>
> **Whether retrainining with SNOWS improves performance**
>
> Yes, we observe benefits to retraining even after pruning with SNOWS. Below we provide a new baseline (FALCON [1]) and include metrics for retraining with unstructured sparsity on on CIFAR-10 and ImageNet. We train for 90 epochs on CIFAR-10 and 5 epochs on ImageNet-1k. Retraining consistently improves performance across all sparsity levels. For example, on CIFAR-10 at 80\% sparsity, retraining improves accuracy drop from -4.74\% to just -1.50\%.
>
> Note: for the retraining experiments, we retrain on the full ImageNet-1k dataset, as opposed to the sample of 3k samples from MiniImageNet-1k we used in our initial experiments.
>
> # ResNet20 on CIFAR10
>
> | **Sparsity** | **MP**   | **WF**   | **CBS**  | **CHITA** | **FALCON** | **FALCON+RT** | **SNOWS**  | **SNOWS+RT** |
> |--------------|----------|----------|----------|-----------|------------|---------------|------------|--------------|
> | 50%          | -2.92    | -1.13    | -0.78    | -0.76     | -0.49      | -0.29         | -0.28      | **-0.24**    |
> | 60%          | -6.12    | -3.40    | -2.48    | -2.14     | -1.69      | -0.78         | -0.66      | **-0.43**    |
> | 70%          | -12.57   | -10.31   | -9.52    | -7.24     | -6.94      | -1.72         | -1.69      | **-0.71**    |
> | 80%          | -37.35   | -28.73   | -40.08   | -33.46    | -26.19     | -3.77         | -4.74      | **-1.50**    |
> | 90%          | -79.57   | -79.87   | -77.68   | -75.76    | -72.22     | -9.76         | -27.76     | **-4.87**    |
>
> # MobileNetV1 on ImageNet-1K
>
> | **Sparsity** | **MP**   | **WF**   | **CBS**  | **CHITA** | **FALCON** | **FALCON+RT** | **SNOWS**  | **SNOWS+RT** |
> |--------------|----------|----------|----------|-----------|------------|---------------|------------|--------------|
> | 50%          | -9.34    | -3.04    | -1.74    | -1.53     | -1.60      | -1.29         | -0.79      | **-0.23**    |
> | 60%          | -30.01   | -11.05   | -5.58    | -4.65     | -4.77      | -3.06         | -2.19      | **-0.79**    |
> | 70%          | -65.17   | -42.59   | -16.84   | -12.55    | -13.55     | -7.21         | -5.61      | **-2.13**    |
> | 80%          | -71.84   | -71.71   | -55.57   | -42.17    | -46.13     | -18.11        | -16.06     | **-6.19**    |
>
> [1] Meng, X., Chen, W., Benbaki, R., & Mazumder, R. (2024). FALCON: FLOP-Aware Combinatorial Optimization for Neural Network Pruning. In Proceedings of The 27th International Conference on Artificial Intelligence and Statistics

---

> > ### Comment · Reviewer_TGxm · 2024-11-27
> > **Post-rebuttal comments**
> >
> > Thank you for the detailed response. The experiments clearly show the benefit of doing Hessian-free optimization. I would recommend to include them in the supplementary. I was already positive (8) and I'll retain my score.

---

> > > ### Author Response · Authors · 2024-11-27
> > >
> > > Dear Reviewer TGxm,
> > >
> > > Thank you for again for your constructive review. We also thought the experiments comparing to Fisher approximation we're of benefit to the paper and we decided to include them in the revised manuscript (see Figure 9). Many thanks to the reviewer for this suggestion and for the positive feedback on our paper.
> > >
> > > Best regards,
> > >
> > > Authors

---

### Author Response · Authors · 2024-11-22
**Summary of Revision Changes**

We would like to thank the reviewers for their detailed feedback and comments. We have provided a revised manuscript incorporating your feedback. We detail the changes below.

**Formatting and organization**

We thank the reviewers for their helpful formatting and structuring suggestions. We have made your formatting changes such as equation references and figure legends, as well as fixing typos. Moreover, we have significantly shortened Section 4 on the SNOWS method, and in its place we have added references to additional experiments (such as run-time and retraining experiments) in Section 5, and added a new Section 6 in the main text on the limitations and conclusions of our work. Moreover, due to the additional content, we have improved the structure of the supplementary section, dividing it into "Primary Methods and Analysis", "Ablation Studies", and "Implementation and Experimental Details".

**Comparison to Fisher approximation**

We have provided new experiments demonstrating the utility of incorporating the true Hessian information versus the popular Fisher approximation. We observe that the second-order descent with the Fisher approximation can take a long time to converge. For large networks, this becomes prohibitive, whereas our Hessian-free method typically can be safely terminated after a few seconds per layer. This is discussed in Section A.2.2.

**Analysis on the SNOWS loss function**

We have provided some additional analysis on the merit of the SNOWS loss function in Section A.1.2, comparing to the regular layer-wise approach, at the request of Reviewer EtDe. To summarize, SNOWS better preserves the input to future layers, during the pruning of a given layer. We thank the reviewer for prompting this further analysis with their question.

**Effect of the parameter $K$ on run-time**

Several reviewers asked for ablations showing the effect of the parameter $K$ on the run-time and memory consumption of our algorithm. This is discussed in the new Section A.2.4. As we mentioned in the main text, the parameter $K$ induces a trade-off between run-time and performance, with larger values of $K$ performing better but being slower to run. In Section A.3.7, we provide some additional run-time experiments comparing to CHITA [1], showing that for moderate-sized $K$, SNOWS performs better and scales much better in terms of architecture size. We reiterate that our method avoids computing, approximating or storing the full network Hessian, and even avoids storing layer-wise Hessians, which makes its complexity much more manageable with respect to architecture size.

**Updating benchmarking**

We have added FALCON, a new benchmark from 2024 AISTATS, to our unstructured sparsity benchmarks (Figure 3). We decided not to include the experiments from [3] in our final manuscript, though we addressed this concern with reviewer 96mw, and we hope our response to the reviewer clarifies this issue. Similarly, to Reviewer YAxu, while we see LayerMerge [4] as a good structured pruning technique, it should not be considered a one-shot pruning method, since the method has close to 10\% accuracy following pruning (cf. Figure 3 of their paper), and requires extensive retraining to recover performance (they use 180 epochs of retraining on ImageNet-1k).

**Retraining experiments**

At the request of several reviewers, we provide experiments in Section A.3.4 showing SNOWS can benefit from even limited retraining (e.g. 5 epochs on ImageNet-1k). We compare to retraining performance of FALCON [2].

**Latency analysis**

A challenge in our evaluation has been the measurement of inference time speed-ups for N:M sparsity patterns. Unfortunately, there is limited native support in PyTorch for such measurements. Current hardware acceleration is particularly constrained, with only the 2:4 sparsity pattern having support through NVIDIA's sparse tensor cores. To address these limitations, we have report comprehensive end-to-end FLOPs reduction across several of our tested architectures and sparsity patterns in Section A.3.6. We tested this end-to-end to ensure our measurements account for all network operations and layers.

[1] Benbaki, R., Chen, W., Meng, X., Hazimeh, H., Ponomareva, N., Zhao, Z., & Mazumder, R. (2023). Fast as CHITA: Neural Network Pruning with Combinatorial Optimization. In Proceedings of the 40th International Conference on Machine Learning.

[2] Meng, X., Chen, W., Benbaki, R., & Mazumder, R. (2024). FALCON: FLOP-Aware Combinatorial Optimization for Neural Network Pruning. In Proceedings of The 27th International Conference on Artificial Intelligence and Statistics (AISTATS).

[3] Hoffmann, J., Agnihotri, S., Saikia, T., & Brox, T. (2021). Towards improving robustness of compressed CNNs. In Proceedings of the ICML Workshop on Uncertainty and Robustness in Deep Learning.

[4] LayerMerge: Neural Network Depth Compression through Layer Pruning and Merging, International Conference on Machine Learning, 2024.

---

### Meta-Review · Area_Chair_aruP · 2024-12-21

**Metareview:**

The paper proposes a one-shot post-training pruning method, that involves layer-wise pruning within a global  optimization. It is an adaptation of previous second order methods which, however, avoids the calculation of the full Hessian by using Hessian-free conjugate gradients.
The proposed method, SNOWS, is interesting and presented within a theoretically sound framework and provides good results that support the theoretical claims. While there are slight remaining concerns on the organization of the paper, the reviewers are overall positive, stressing the theoretic contribution of the work.

**Additional Comments On Reviewer Discussion:**

The paper initially had one positive reviewer (TGxm), asking for some clarifications, who maintained the positive score after the rebuttal. Reviewers (XYMY) and (96mw) were initially quite critical with issues raised in the paper writing and concept definitions. After the rebuttal/revision, the scores were raised to 5 and 8 respectively, indicating that all crucial concerns were addressed.

---

### Decision · Program_Chairs · 2025-01-22

Accept (Poster)